# Functional instability allows access to DNA in longer transcription Activator-Like effector (TALE) arrays

**Kathryn Geiger-Schuller[1,2†], Jaba Mitra[3], Taekjip Ha[1,2,4,5,6,7,8], Doug Barrick[1,2]\***

[1]T.C. Jenkins Department of Biophysics, Johns Hopkins University, Baltimore, United States; [2]Program in Molecular Biophysics, Johns Hopkins University, Baltimore, United States; [3]Materials Science and Engineering, University of Illinois Urbana-Champaign, Urbana, United States; [4]Department of Physics, Center for the Physics of Living Cells, University of Illinois at Urbana Champaign, Urbana, United States; [5]Institute for Genomic Biology, University of Illinois at Urbana Champaign, Urbana, United States; [6]Department of Biomedical Engineering, Johns Hopkins University, Baltimore, United States; [7]Department of Biophysics and Biophysical Chemistry, Johns Hopkins University, Baltimore, United States; [8]Howard Hughes Medical Institute, Baltimore, United States

**Abstract** Transcription activator-like effectors (TALEs) bind DNA through an array of tandem 34-residue repeats. How TALE repeat domains wrap around DNA, often extending more than 1.5 helical turns, without using external energy is not well understood. Here, we examine the kinetics of DNA binding of TALE arrays with varying numbers of identical repeats. Single molecule fluorescence analysis and deterministic modeling reveal conformational heterogeneity in both the free- and DNA-bound TALE arrays. Our findings, combined with previously identified partly folded states, indicate a TALE instability that is functionally important for DNA binding. For TALEs forming less than one superhelical turn around DNA, partly folded states inhibit DNA binding. In contrast, for TALEs forming more than one turn, partly folded states facilitate DNA binding, demonstrating a mode of 'functional instability' that facilitates macromolecular assembly. Increasing repeat number slows down interconversion between the various DNA-free and DNA-bound states.
DOI: https://doi.org/10.7554/eLife.38298.001

**\*For correspondence:**
barrick@jhu.edu

**Present address:** [†]Broad Institute of Harvard and Massachusetts Institute of Technology, Cambridge, United States

## Introduction

Transcription activator-like effectors (TALEs) are bacterial proteins containing a domain of tandem DNA-binding repeats as well as a eukaryotic transcriptional activation domain (*Kay et al., 2007*; *Römer et al., 2007*). The repeat domain binds double stranded DNA with a register of one repeat per base pair. Specificity is determined by the sequence identity at positions twelve and thirteen in each TALE repeat, which are referred to as repeat variable diresidues (RVDs) (*Boch et al., 2009*; *Miller et al., 2015*; *Moscou and Bogdanove, 2009*). This specificity code has enabled design of TALE-based tools for transcriptional control (*Cong et al., 2012*; *Geissler et al., 2011*; *Li et al., 2012*; *Mahfouz et al., 2012*; *Morbitzer et al., 2010*; *Zhang et al., 2011*), DNA modifications (*Maeder et al., 2013*), in-cell microscopy (*Ma et al., 2013*; *Miyanari et al., 2013*), and genome editing (TALENs) (*Christian et al., 2010*; *Li et al., 2011*).

TALE repeat domains wrap around DNA in a continuous superhelix of 11.5 TALE repeats per turn (*Deng et al., 2012*; *Mak et al., 2012*). Because TALEs contain on average 17.5 repeats (*Boch and Bonas, 2010*), most form over 1.5 full turns around DNA. Many multisubunit proteins that form rings around DNA require energy in the form of ATP to open or close around DNA (reviewed in

**eLife digest** The DNA contains all the information needed to build an organism. It is made up of two strands that wind around each other like a twisted ladder to form the double helix. The strands consist of sugar and phosphate molecules, which attach to one of for bases. Genes are built from DNA, and contain specific sequences of these bases. Being able to modify DNA by deleting, inserting or changing certain sequences allows researchers to engineer tissues or even organisms for therapeutical and practical applications.

One of these gene editing tools is the so-called transcription activator-like effector protein (or TALE for short). TALE proteins are derived from bacteria and are built from simple repeating units that can be linked to form a string-like structure. They have been found to be unstable proteins. To bind to DNA, TALES need to follow the shape of the double helix, adopting a spiral structure, but how exactly TALE proteins thread their way around the DNA is not clear.

To investigate this, Geiger-Schuller et al. monitored single TALE units using fluorescent microscopy. This way, they could exactly measure the time it takes for single TALE proteins to bind and release DNA. The results showed that some TALE proteins bind DNA quickly, whereas others do this slowly. Using a computer model to analyze the different speeds of binding suggested that the fast binding comes from partly unfolded proteins that quickly associate with DNA, and that the slow binding comes from rigid, folded TALE proteins, which have a harder time wrapping around DNA. This suggest that the unstable nature of TALEs, helps these proteins to bind to DNA and turn on genes.

These findings will help to design future TALE-based gene editing tools and also provide more insight into how large molecules can assemble into complex structures. A next step will be to identify TALE repeats with unstable states and to test TALE gene editing tools that have intentionally placed unstable units.

DOI: https://doi.org/10.7554/eLife.38298.002

O'Donnell and Kuriyan, 2006), yet TALEs are capable of wrapping around DNA without energy from nucleotide triphosphate hydrolysis. One possibility is that TALEs bind DNA through an energetically accessible open conformation. Consistent with this possibility, we previously demonstrated that TALE arrays can populate partly folded and broken states (*Geiger-Schuller and Barrick, 2016*). By measuring the length-dependence of protein stability and employing a statistical mechanics Ising model, we previously described several different TALE partly folded states termed 'end-frayed', 'internally unfolded', and 'interfacially fractured' states. Although the calculated populations of partly folded states in TALE repeat arrays are small, they are many orders of magnitude larger than populations of partly folded states in other previously studied repeat arrays (consensus ankyrin [*Aksel et al., 2011*] and DHR proteins [*Geiger-Schuller et al., 2018*]) suggesting a potential functional role for the high populations of partially folded states in TALE repeat arrays.

Consensus TALEs (cTALEs) are designed homopolymeric arrays composed of the most commonly observed residue at each of the 34 positions of the repeat (*Geiger-Schuller and Barrick, 2016*). In addition to simplifying analysis of folding and conformational heterogeneity in this study, the consensus approach simplifies analysis of DNA binding, eliminating contributions from sequence heterogeneity and providing an easy means of site-specific labeling.

Here we characterize DNA binding kinetics of cTALEs using total internal reflection fluorescence single-molecule microscopy. We find that consensus TALE arrays bind to DNA reversibly, with high affinity. Analysis of the dwell-times of the on- and off-states reveals multiphasic binding and unbinding kinetics, suggesting conformational heterogeneity in both the free and DNA bound state. We develop a deterministic optimization analysis that supports such a model, and provides rate constants for conformational changes in the unbound and bound states, and rate constants for binding and dissociation. Comparing the dynamics observed here to previously characterized local unfolding suggests that locally unfolded states inhibit binding of short cTALE arrays (less than one full superhelical turn around DNA), whereas they promote binding of long arrays (more than one full superhelical turn). Whereas local folding of transcription factors upon DNA binding is well documented (*Spolar and Record, 1994*), local unfolding in the binding process is not. Our results present a new

mode of transcription factor binding where the major conformer in the unbound state is fully folded, requiring partial unfolding prior to binding. The critical role of such high energy partly folded states is an exciting example of 'functional instability', in which formation of a functional complex is impeded by the fully folded native state, and is instead facilitated by partial disruption of native structure.

## Results

### cTALE design

Consensus TALE (cTALE) repeat sequence design was described previously (*Geiger-Schuller and Barrick, 2016*). To avoid self- association of cTALE arrays, we fused arrays to a conserved N-terminal extension of the PthXo1 gene. Although the sequence of this domain shows little similarity to TALE repeat sequences, the structure of this domain closely mimics four TALE repeats (*Gao et al., 2012*; *Mak et al., 2012*) and is required for binding and full transcriptional activation (*Gao et al., 2012*). In this study, all repeat arrays contain this solubilizing N-terminal domain.

### cTALE local instability promotes population of partly folded states

In a previous study of the folding of a series of cTALE arrays, we used a nearest-neighbor Ising model to determine energies of intrinsic repeat folding and interfacial interaction between repeats. This analysis allows us to quantify the energies of different partly folded states. *Figure 1A* depicts three types of partly folded states of a generic repeat protein. In the fully folded state, all repeats are folded, and all interfaces are intact. In the end-frayed states, one or more terminal repeats are

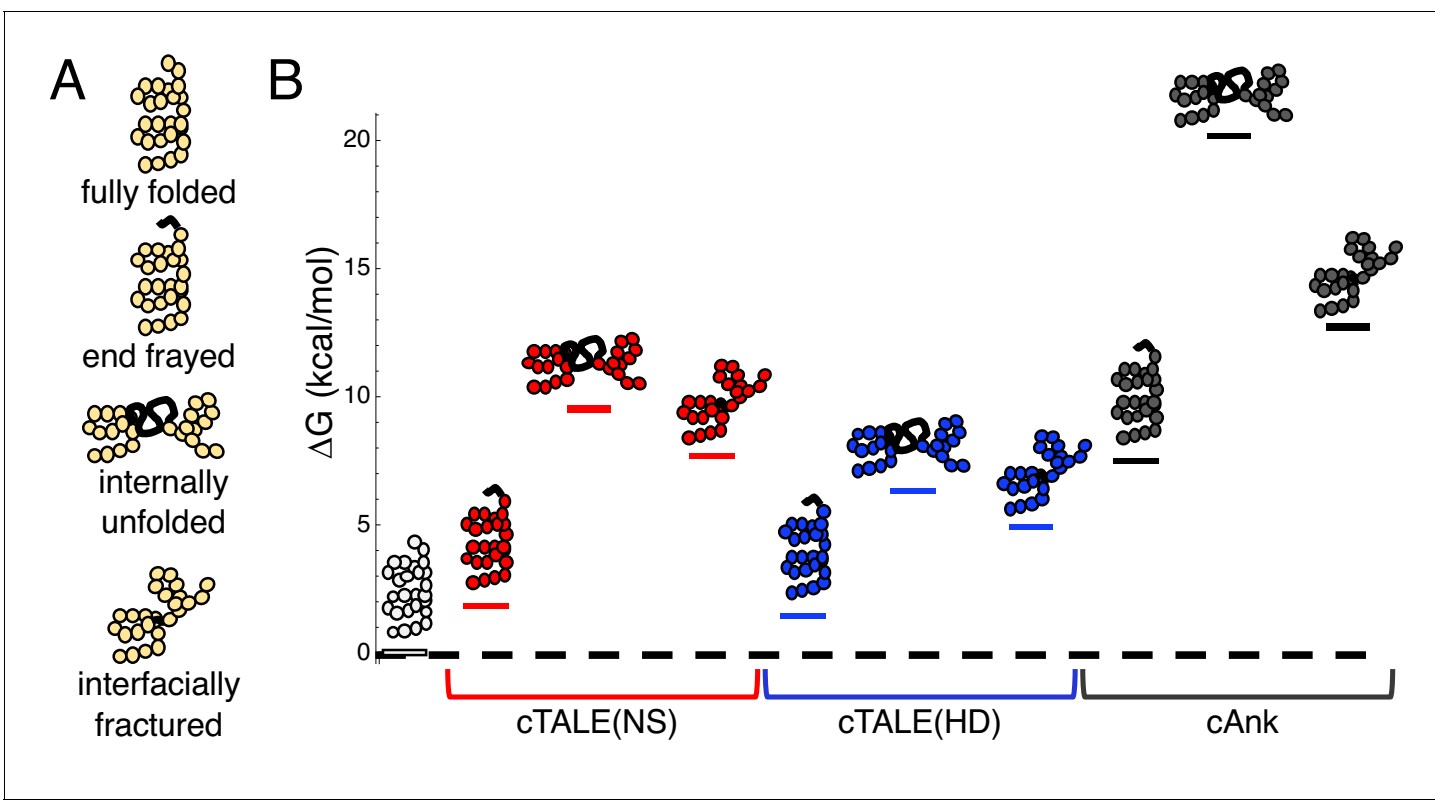

**Figure 1.** cTALEs populate partly folded states. (**A**) Cartoon of different partly folded TALE conformational states. End-frayed states have one (or more) terminal repeats unfolded. Internally unfolded states have a central repeat unfolded. Interfacially fractured states have a disrupted interface between adjacent repeats. (**B**) Free energies of partly folded states, calculated from previously published measurements and analysis (*Geiger-Schuller and Barrick, 2016*), relative to the fully folded state, for consensus TALE repeats with the NS repeat-variable diresidue sequence (cTALE(NS), red), consensus TALE repeats with the HD repeat-variable diresidue sequence (cTALE(HD), blue), and consensus ankyrin repeats (cAnk, black).
DOI: https://doi.org/10.7554/eLife.38298.003

unfolded but all interfaces (except the interface between the unfolded and adjacent folded repeat), are intact. In the internally unfolded states, a central repeat is unfolded but all interfaces (except the interfaces involving the unfolded repeats) are intact. In the interfacially ruptured state, all repeats are folded but one interface is disrupted due to local structural distortion.

*Figure 1B* shows calculated free energy differences between various partly folded states and the fully folded repeat array for two different RVDs (NS and HD) in an otherwise identical consensus sequence background, using the intrinsic and interfacial engeries we determined previously (*Geiger-Schuller and Barrick, 2016*). The distribution of partly folded states is calculated for different 20-repeat arrays containing two types of TALE arrays (with the NS RVD in red and with the HD RVD in blue) as well as consensus ankyrin arrays (cAnk in black; Materials and methods). For cTALE arrays, end frayed states are within a few $k_BT$ of the folded state, internally unfolded states are highest in energy, and interfacially ruptured states fall energetically between end frayed and internally unfolded states.

Changing the RVD sequence affects the distribution of these partly folded states: arrays containing HD repeats are more likely to internally unfold or interfacially rupture than arrays containing NS repeats. However, both types of cTALEs are more likely to populate many of these partly folded states than cAnk is to populate even the lowest energy partly folded state, the end frayed state. Thus, compared to ankyrin repeats, cTALEs are locally unstable, meaning they are likely to form partly folded states. As these states disrupt the superhelix, they may facilitate DNA binding.

## Single-molecule studies of cTALE binding to DNA

To ask if cTALE local instability is relevant for DNA binding kinetics, DNA binding trajectories were measured using single molecule total internal reflection fluorescence (smTIRF). *Figure 2A* shows a schematic of the smTIRF experiments performed to measure DNA binding. For site-specific cTALE labeling, R30 is mutated to cysteine in a single repeat. Position 30 is frequently a cysteine in naturally occurring TALEs (in earlier folding studies, arginine was chosen in the consensus sequence to avoid disulfide formation; *Geiger-Schuller and Barrick, 2016*). This cysteine was Cy3 (FRET donor)-labelled using maleimide chemistry, and the Cy3-lablelled TALE array was attached to biotinylated slides via the C-terminal His$_6$ tag and $\alpha$-Penta•His antibodies. At salt concentrations below 300 mM NaCl, cTALEs aggregate. Because DNA binding is weak at high salt concentrations, measuring binding kinetics in bulk at high salt is not possible. However, tethering cTALEs to the quartz slide at high salt prevents self-association, even at the low salt concentrations required to study DNA binding kinetics. A histogram of NcTALE$_8$ (8 NS-type repeats and the N-terminal domain) labeled via a cysteine in the first repeat shows a single peak at zero FRET efficiency, as expected for donor-only constructs (*Figure 2B*).

To test for DNA binding to tethered cTALE constructs, we added 5'-Cy5 (FRET acceptor)-labeled 15 bp-long DNA (Cy5.A$_{15}$/T$_{15}$) to tethered NcTALE$_8$. This results in a new peak at a FRET efficiency of 0.45, indicating that DNA binds directly to cTALE arrays. As DNA concentration in solution is increased, the peak at 0.45 FRET efficiency increases in population (*Figure 2C–D*), suggesting a measurable equilibrium between free and bound DNA rather than saturation or irreversible binding. In support of this, single molecule time trajectories show interconversion between bound and unbound states, providing access to rates of binding and dissociation.

As expected for reversible complex formation, the peak at 0.45 FRET efficiency can be competed away by adding mixtures of labeled and unlabeled DNA to pre-formed cTALE-labelled DNA complexes (schematic shown in *Figure 2E*; pre-formed complex shown in *Figure 2F*, competition data shown in *Figure 2G–H*). Challenging pre-formed complexes with a mixture of 5 nM unlabeled DNA and 15 nM labeled DNA results in a slight decrease in the population of the peak at 0.45 FRET (compare *Figure 2F and G*). Challenging with a mixture of 50 nM unlabeled DNA and 15 nM labeled DNA further decreases the peak at 0.45 FRET (*Figure 2H*).

To test whether tethering of cTALE arrays impacts DNA binding, we also performed experiments with tethered dsDNA and free Cy3-labeled NcTALE$_8$. Here, we tethered biotinylated, Cy5 (FRET acceptor)-labeled 15 bp-long DNA (Cy5.A$_{15}$/biotin.T$_{15}$). Addition of Cy3-labeled NcTALE$_8$ at high salt concentration results in a new peak at a FRET efficiency of 0.5 (*Figure 2—figure supplement 1*). These FRET distributions are similar to those obtained from tethering cTALEs to the surface and adding free dsDNA, suggesting that the interaction of cTALE arrays with dsDNA is not significantly impacted by surface immobilization. Since cTALEs tend to associate at physiological salt

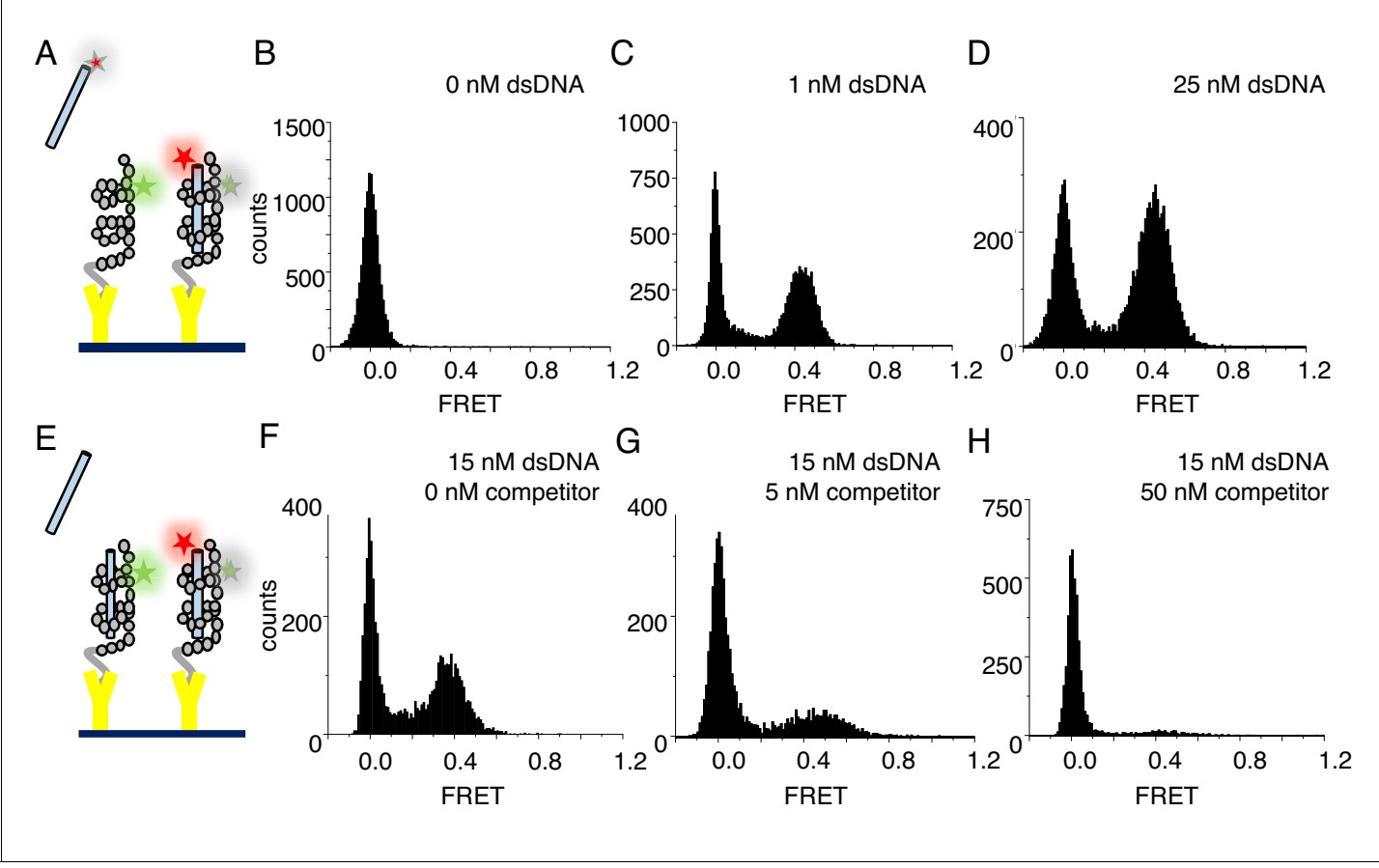

**Figure 2.** cTALEs bind dsDNA reversibly. (A) Schematic of single-molecule FRET assay, with donor-labelled cTALE attached to a surface, and acceptor-labelled DNA free in solution. (B–D) Single molecule FRET histograms show the appearance of a peak at a FRET efficiency of 0.45 with increasing labelled DNA, consistent with a DNA-bound cTALE. (E) Schematic of single-molecule FRET competition assay, with donor-labelled cTALE attached to a surface, and acceptor-labelled DNA as well as competitor unlabeled DNA free in solution. (F–H) Single molecule FRET histograms show the disappearance of the peak at 0.45 FRET efficiency with increasing unlabeled competitor DNA. Conditions: 20 mM Tris pH 8.0, 200 mM KCl.

DOI: https://doi.org/10.7554/eLife.38298.004

The following figure supplement is available for figure 2:

**Figure supplement 1.** Untethered cTALEs bind dsDNA.

DOI: https://doi.org/10.7554/eLife.38298.005

concentration, we used the format where cTALEs were tethered to the slide and incubated with freely diffusing dsDNA for all subsequent experiments.

## cTALE arrays display multiphasic DNA-binding and -unbinding kinetics

In addition to the short smTIRF movies used to generate smFRET histograms from many molecules, long movies were also collected to examine the extended transitions of individual molecules between the low- and high-FRET (0 and 0.45) (*Figure 3A–B*). A transition from low to high FRET (0 to 0.45) indicates that the acceptor fluorophore on DNA moved close enough to the donor on the protein for FRET and is likely a binding event. A transition from high to low FRET (0.45 to 0.0) indicates the acceptor fluorophore on DNA moved too far away from the donor on the protein for FRET and is likely an unbinding event. Low-FRET states show low colocalization with signal upon direct excitation of the acceptor, confirming that high-FRET states are DNA-bound states and low-FRET states are DNA-free states (*Figure 3—figure supplement 1*). These long single molecule traces show both long- and short-lived low- and high-FRET states, indicating that kinetics are multi-phasic (*Figure 3A–B*). Binding events (transitions from low to high FRET) become more frequent as bulk DNA concentration increases (compare representative traces at 1 nM dsDNA to 15 nM dsDNA;

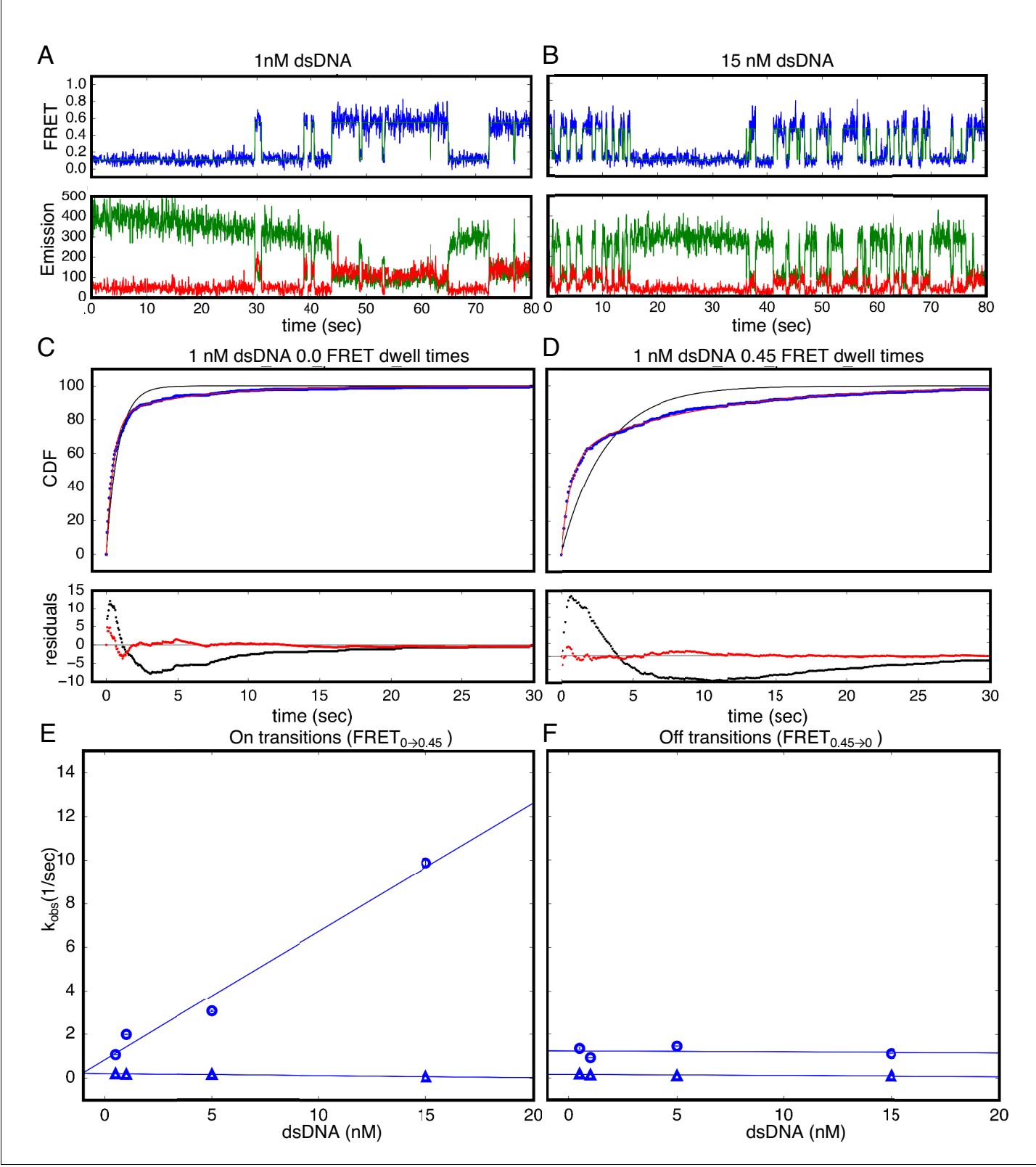

**Figure 3.** Single Molecule kinetics show multiple phases in binding and unbinding kinetics for NcTALE$_8$ binding to dA$_{15}$/T$_{15}$ duplex DNA. (A–B) Long time trajectories showing transitions between low- and high-FRET states (efficiencies of 0 and 0.45). The top panel shows calculated FRET efficiency in blue and two-state Hidden Markov Model fit in green (*McKinney et al., 2006*). The bottom panels show Cy3 and Cy5 fluorescence emission in green and red respectively. At low DNA concentration (A), the low FRET state predominates. As DNA concentration is increased (B), more time is spent in the

*Figure 3 continued on next page*

*Figure 3 continued*

high FRET state, because the dwell times in the low FRET state are shorter. At low DNA concentrations, there appears to be long- and short-lived high-FRET states. Likewise, at near-saturating DNA concentrations, there appear to be long and short-lived low FRET states. (C, D) Cumulative distributions of low- and high-FRET dwell times (blue circles). Fits to single-exponentials (black) show large nonrandom residuals (lower panels), consistent with the heterogeneity noted in (A) and (B). Double-exponentials (red) give smaller, more uniform residuals. (E) Apparent association rate constants as a function of DNA concentration. The apparent rate constant for the fast phase depends on DNA concentration (blue circles), indicating a bimolecular step binding event. The apparent rate constant for the slow phase does not depend on DNA-concentration (blue triangles), suggesting an isomerization event. (F) Apparent dissociation rate constants as a function of DNA concentration (phase one shown in blue circles, and phase two shown in blue triangles). Neither phase shows a DNA concentration dependence, indicating a dissociation and/or isomerization events. 68% confidence intervals are estimated using the conf_interval function of lmfit by performing F-tests (*Newville et al., 2014*). Conditions: 20 mM Tris pH 8.0, 200 mM KCl.

DOI: https://doi.org/10.7554/eLife.38298.006

The following source data and figure supplements are available for figure 3:

**Source data 1.** List of values used to construct long time trajectories displayed in *Figure 3A*.
DOI: https://doi.org/10.7554/eLife.38298.009
**Source data 2.** List of values used to construct long time trajectories displayed in *Figure 3B*.
DOI: https://doi.org/10.7554/eLife.38298.010
**Source data 3.** List of values used to construct CDFs displaed in *Figure 3C*.
DOI: https://doi.org/10.7554/eLife.38298.011
**Source data 4.** List of values used to construct CDFs displayed in *Figure 3D*.
DOI: https://doi.org/10.7554/eLife.38298.012
**Figure supplement 1.** Alternating laser experiments show agreement between cTALE$_8$ FRET and colocalization kinetics.
DOI: https://doi.org/10.7554/eLife.38298.007
**Figure supplement 2.** cTALEs do not slide onto ends of short dsDNA.
DOI: https://doi.org/10.7554/eLife.38298.008

*Figure 3A* and *Figure 3B*). Cumulative distributions generated from dwell times in the low FRET state at a given DNA concentration are best-fit by a double-exponential decay, indicating a minimum of two kinetic phases associated with binding events (*Figure 3C*). Cumulative distributions generated from dwell times in the high FRET state are also best-fit by a double-exponential decay, indicating that there are a minimum of two kinetic phases for unbinding as well (*Figure 3D*).

The rate constant for the fast phase in DNA binding shows a linear increase with DNA concentration (*Figure 3E*), indicating that this step involves an associative binding mechanism. The slope of the rate constant for the fast phase as a function of DNA concentration gives a bimolecular rate constant of $5.9 \times 10^8$ nM$^{-1}$s$^{-1}$, close to the diffusion limit. The rate constant for the slower phase (0.14 s$^{-1}$) is independent of DNA concentration indicating a unimolecular isomerization mechanism (*Figure 3E*).

In contrast, neither of the two fitted rate constants for transitions from high to low FRET (0.45 to 0.0; unbinding events) depends on DNA concentration, suggesting that unbinding involves two (or more) unimolecular processes (*Figure 3F*). The rate constants of these two phases are 1.2 s$^{-1}$ and 0.13 s$^{-1}$ respectively.

To rule out kinetic contributions of TALEs threading axially onto the ends of short DNAs, binding kinetics were measured with capped double-helical DNA sites. Capped DNA was generated by forming 5'digoxygenin-A$_5$-Cy5-A$_{15}$ duplexed with 5'-digoxygenin-T$_{26}$ and adding a three-fold molar excess of anti-Digoxygenin. Low and high FRET dwell time cumulative distributions generated from capped DNA-binding kinetics are bi-phasic, similar to distributions from uncapped DNA (*Figure 3—figure supplement 2*). The DNA concentration-independent rate constant for binding is roughly the same for capped DNA as for uncapped DNA (compare FRET$_{L \to H}$ red and blue triangles in *Figure 3—figure supplement 2*), as are the dissociation rate constants (compare FRET$_{H \to L}$ red and blue triangles in as well as FRET$_{H \to L}$ red and blue circles in *Figure 3—figure supplement 2*). The rate constant for bimolecular binding of capped DNA decreases compared to that for uncapped DNA (compare FRET$_{L \to H}$ red and blue circles in *Figure 3—figure supplement 2*), which is consistent with the expected decrease in the rate of diffusion of the larger capped DNA. To assess the effect of molecular weight increase on diffusion of capped versus uncapped DNA, Sednterp (*Laue et al., 1992*), a program commonly used to estimate sedimentation and diffusion properties of biomolecules, was used to estimate maximum diffusion coefficients. Including the two antibodies bound on the ends of

capped DNA (320 kDa total) gives an estimated diffusion coefficient of $4.7 \times 10^{-7}$ cm$^2$s$^{-1}$, which is much lower than the estimated diffusion coefficient of the uncapped DNA ($1.5 \times 10^{-6}$ cm$^2$s$^{-1}$). This ~3.6 fold decrease in the diffusion constant for capped DNA is similar to the 6.7-fold decrease in the bimolecular rate constant for binding of capped DNA (*Figure 3—figure supplement 2*).

## Modifying the dsDNA sequence to include an anchoring 5' T impacts unbinding kinetics

The heterogeneity in the DNA-bound state may either result from conformational heterogeneity of the bound cTALE array, or from heterogeneity in the registry between the cTALE array and the DNA. Because there are three more base pairs than TALE repeats, there are several available binding registers where all TALE repeats are bound to DNA; ignoring end-effects, these registers are expected to have similar energetics. Although variation in registry would not be expected for natural TALE arrays that bind to high-complexity DNA sequences, it is more likely for the simple poly-A sequence here. To test whether bound-state heterogeneity results from a variation in cTALE-DNA registry, we altered the dsDNA sequence to promote a specific bound state. Previous studies indicate that the addition of a 5' T to the binding sequence, referred to as a T-anchored binding site, greatly enhances TALE binding activity (*Boch et al., 2009*) depending also on the presence of certain RVDs and degree of mismatch relative to cognate DNA (*Schreiber and Bonas, 2014*). To monitor kinetics with a T-anchored DNA, we added Cy5-labeled 15 bp-long DNA (Cy5.TA$_{14}$/T$_{14}$A) to tethered NcTALE$_8$. Similar to the homopolymeric DNA (Cy5.A$_{15}$/T$_{15}$), this results in a peak at a FRET efficiency of 0.55, which increases with increasing DNA concentration (*Figure 4A,B*).

To examine binding and unbinding kinetics of cTALEs interacting with T-anchored DNA, long movies were recorded to visualize multiple transitions of individual molecules between the low- and high-FRET states (*Figure 4A–B*). As with A$_{15}$/T$_{15}$ DNA, these long single molecule traces show both long- and short-lived low-FRET states, consistent with multiphasic binding kinetics. In contrast, dissociation of T-anchored DNA only shows long-lived high-FRET states, suggesting a single kinetic phase for unbinding (*Figure 4A–B*) and cumulative distributions generated from dwell times in the high FRET state are well-fitted by a single-exponential decay (*Figure 4D*).

As with the binding kinetics measured for the homopolymeric DNA, the rate constant for the fast phase in T-anchored DNA binding shows a linear increase with DNA concentration (*Figure 4E*). The bimolecular rate constant for this phase ($3.9 \times 10^8$ nM$^{-1}$s$^{-1}$) is close to the diffusion limit, as was seen for homopolymeric DNA ($5.9 \times 10^8$ nM$^{-1}$s$^{-1}$). The rate constant for the slower phase (0.34 s$^{-1}$) is also similar to the homopolymeric DNA rate constant for the slower phase (0.14 s$^{-1}$, *Figure 4E*).

As with the unbinding kinetics measured for the homopolymeric DNA, the fitted rate constant for transition from high to low FRET (0.55 to 0.0; unbinding events) does not depend on DNA concentration (*Figure 4F*). Compared to the unbinding kinetics measured for the homopolymeric DNA, the anchored unbinding cumulative distributions are well-fitted by a single exponential (although the model with bound-state heterogeneity still fits slightly better, as shown below). There are two possible interpretations of this result. Either the T-anchored DNA impacts the binding mechanism such that unbinding involves one simple unimolecular processes, or the T-anchored DNA shifts the microscopic rate constants such that, although unbinding involves two (or more) unimolecular processes, the amplitudes are very different, or apparent rates are too close to resolve them. Either way, the large effect of T-anchor on unbinding kinetics supports the idea that bound-state heterogeneity results from variation in the the registry between the cTALE array and the DNA.

## Longer cTALEs have slower DNA binding and unbinding kinetics

To examine how increasing the length of the cTALE array influences DNA binding, we generated Cy3-labelled constructs with 16 and 12 cTALE repeats, and measured binding to a longer Cy5-labelled DNA (A$_{23}$/T$_{23}$). We observed a low FRET value near 0.2 for the bound cTALE$_{12}$ state, indicating that the first cTALE12 repeat is farther from the 5'-DNA-bound acceptor fluorophore than in the A$_{15}$/T$_{15}$ DNA complex. Attempts with the 16-repeat cTALE to increase FRET efficiency by moving the position of the mutated cysteine to the fourteenth repeat were unsuccessful. Thus, we used a fluorescence colocalization microscopy protocol to monitor binding of longer cTALE arrays to A$_{23}$/T23 DNA (*Figure 5—figure supplement 1*). In this protocol, Cy3 was first imaged for ten camera frames (1017.5 msec total) to identify positions of single TALE molecules. Then a long time series of

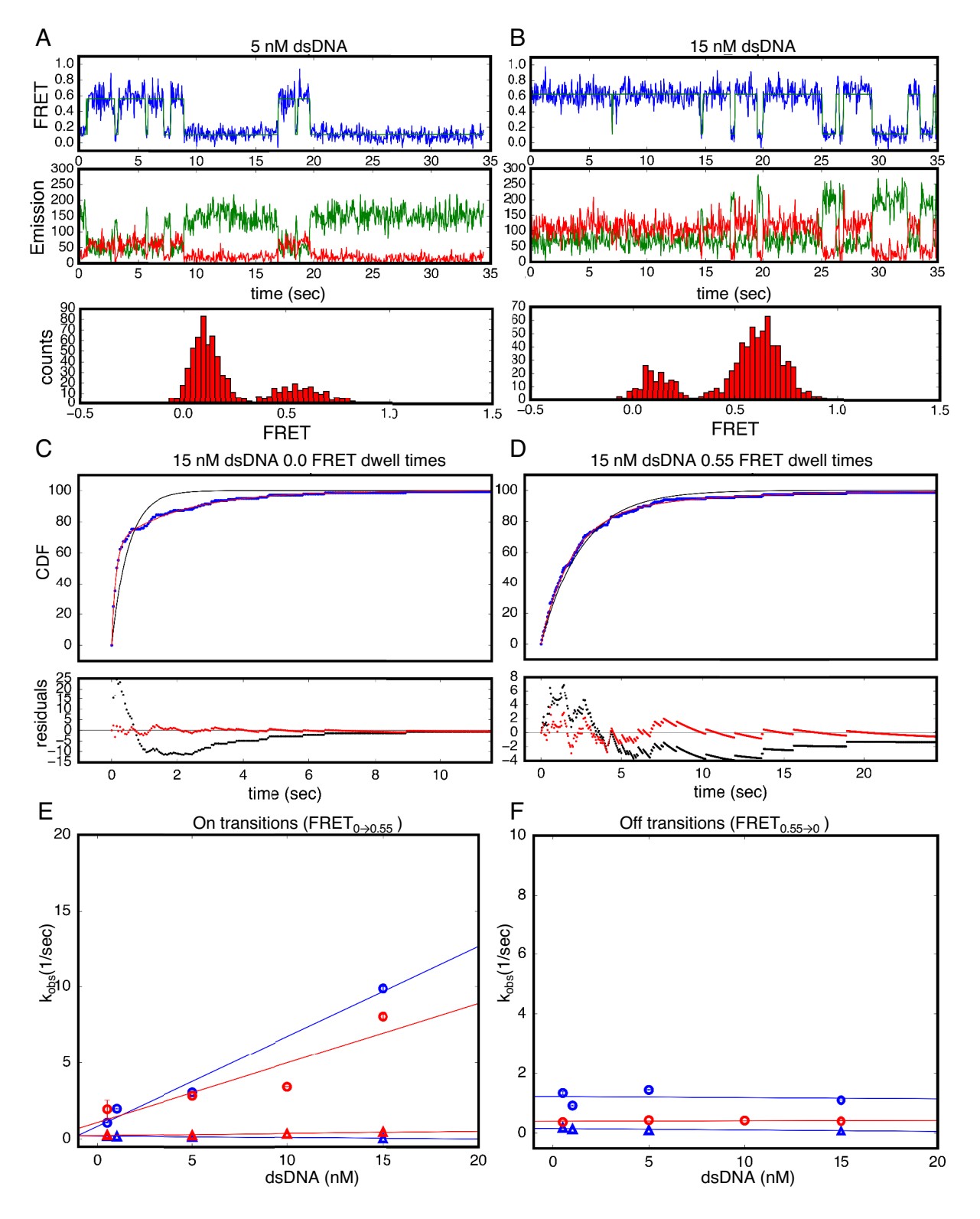

**Figure 4.** Single Molecule kinetics using T-anchored DNA shows a single unbinding phase. (**A–B**) Long time trajectories showing transitions between low- and high-FRET states (efficiencies of 0 and 0.55). The top and middle panels are as in *Figure 3A and B*. The bottom panels show FRET histograms generated using calculated FRET values at each time point in traces A and B. (**C**) Cumulative distributions of low-FRET dwell times (blue circles). A Fit to single-exponential (black) shows large nonrandom residuals (lower panel). (**D**) Cumulative distributions of high-FRET dwell times (blue circles). A fit to

*Figure 4 continued*

single-exponential (black) shows random residuals (lower panel), implying decreased heterogeneity in the T-anchored bound-state. (E) Apparent association rate constants as a function of DNA concentration for the T-anchored $TA_{14}/A_{14}T$ dsDNA (red) and the original $A_{15}/T_{15}$ dsDNA (blue). The apparent rate constants for the fast phase depend on DNA concentration (circles), indicating a bimolecular step binding event. The apparent rate constants for the slow phase do not depend on DNA-concentration (triangles), suggesting an isomerization event. (F) Apparent dissociation rate constants as a function of DNA concentration (T-anchored dSDNA, red circles; original $A_{15}/T_{15}$ dsDNA, blue circles and triangles). 67.4% confidence intervals are estimated using the conf_interval function of lmfit by performing F-tests (*Newville et al., 2014*). Conditions: 20 mM Tris pH 8.0, 200 mM KCl.

DOI: https://doi.org/10.7554/eLife.38298.013

fluorescence images of Cy5 signal were collected through directly exciting Cy5 on the DNA, and time trajectories of Cy5 signal were generated from the initially identified single TALE positions.

Increasing the number of cTALE repeats from 8 to 12 and 16 dramatically affects DNA binding kinetics. Long movies collected over a range of DNA concentrations show short- and long-lived Cy5 signal on and off states, indicating a level of kinetic heterogeneity similar to $NcTALE_8$ (*Figure 5—figure supplement 1*). Single molecule traces were analyzed using a thresholding filter (see Materials and methods and *Figure 5—figure supplement 1*) to identify states and dwell times. Cumulative distributions were generated from dwell times at low Cy5 signal (unbound states, with lifetimes representing binding kinetics), and at high Cy5 signal (bound states, with dwell times representing unbinding kinetics). As with the eight repeat constructs, unbound cumulative distributions for these longer TALE arrays are best-fit by double exponential decays, particularly at high DNA concentrations (compare the cumulative distribution at low DNA concentration, *Figure 5—figure supplement 2A*, to cumulative distribution at 5 nM DNA, *Figure 5—figure supplement 2B*). Bound cumulative distributions for longer TALE arrays are best-fit by double exponential decays (*Figure 5—figure supplement 2C–D*). All apparent rate constants are much smaller for $NcTALE_{16}$ and $NcTALE_{12}$ (green/black circles and triangles, *Figure 5A–B*) compared to $NcTALE_8$, indicating that binding and unbinding is impeded by increasing the length of the binding surface between cTALEs and their cognate DNA (*Figure 5C*). To address whether differences in binding kinetics are related to experimental differences between colocalization and FRET assays, alternating laser experiments were performed by switching between FRET and colocalization detection (every five frames) within single molecule trajectories (*Figure 3—figure supplement 1*). Changes in FRET and colocalization signals occurred simultaneously according to single molecule time traces, showing that differences in binding and unbinding kinetics of short and longer cTALEs are not due to differences in colocalization and FRET assays (*Figure 3—figure supplement 1*).

## A deterministic approach to modeling cTALE-DNA binding kinetics

To determine how the kinetic changes above are partitioned into underlying kinetic steps in binding, we fitted various kinetic models to the cumulative distributions for binding and unbinding. In addition to providing information about the mechanism of binding, this approach allows us to estimate the underlying microscopic rate constants and compare them for different constructs. This approach is generally applicable to studies of complex single molecule kinetics. Numerical integration was used to calculate the relative population of cTALE states as a function of time (*Figure 6A–C and G–I*), given a binding mechanism, an associated set of rate laws, and a set of initial conditions. Cumulative distributions of unbound dwell times represent the distribution of times single molecules spent in the unbound state before transitioning into the bound state, allowing us to split the kinetic scheme when fitting to single-molecule dwell times.

Among the various models tested, the model that is most consistent with the data has two unbound DNA-free states and two DNA-bound states. This is consistent with alternating laser experiments showing that DNA is only colocalized when cTALEs are in the high FRET state (*Figure 3—figure supplement 1*). This four-state model includes a TALE isomerization step in the absence of DNA from a DNA-binding incompetent conformation (which we refer to as TALE) to DNA-binding competent conformation (which we refer to as TALE*). The DNA-binding competent TALE* conformer binds and unbinds DNA (called TALE* when DNA free and TALE*~DNA when DNA-bound). Before unbinding, a fraction of TALE*~DNA isomerizes to a longer-lived DNA-bound state called TALE$^{‡}$~DNA.

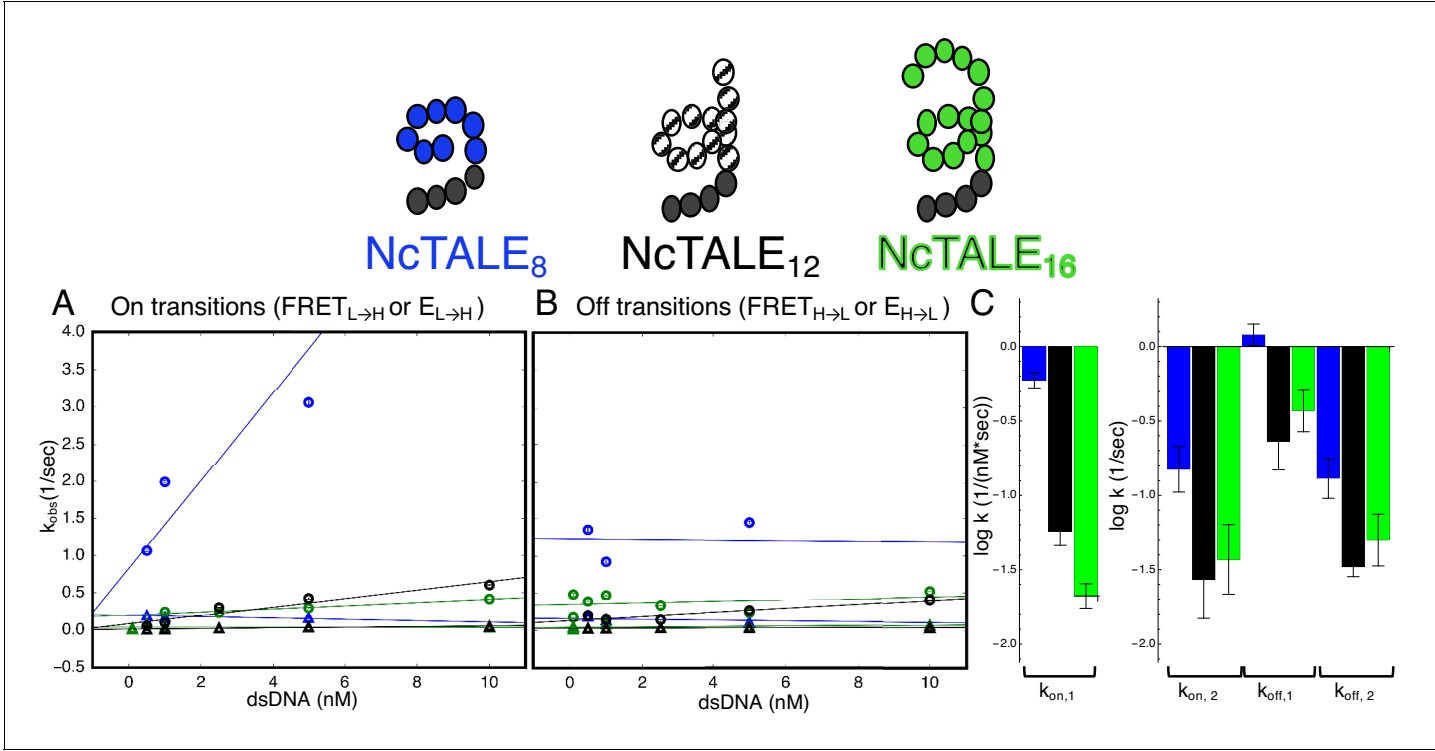

**Figure 5.** A 16-repeat TALE protein binds and unbinds DNA more slowly than an eight repeat protein. (**A**) Apparent association rate constants as a function of DNA concentration for an eight repeat cTALE (blue) binding to $dA_{15}/T_{15}$ duplex DNA, and for a 12 repeat cTALE (black) and a 16 repeat cTALE (green) binding to $dA_{23}/T_{23}$ duplex DNA. eight repeat TALE kinetics are measured by FRET ($FRET_{L \to H}$) while 12 and 16 repeat TALE kinetics are measured by colocalization ($E_{L \to H}$). The apparent rate constants for the fast phase of binding are DNA concentration dependent (blue, black, and green circles), indicating a bimolecular binding event. The DNA concentration-dependence is greatest (larger slope) for the eight repeat cTALE. The apparent rate constants for the slow phase do not depend on DNA-concentration (blue, black, and green triangles), suggesting an isomerization event. (**B**) Apparent dissociation rate constants as a function of DNA concentration (phase one shown in circles, and phase two shown in triangles). Neither phase shows a DNA concentration dependence, indicating a dissociation and/or isomerization events. Rate constants for all phases are slower for the 12-repeat construct (black) and 16-repeat construct (green) than for the 8-repeat construct (blue), particularly for the bimolecular binding step. (**C**) $Log_{10}$ of rate constants for 8 (blue), 12 (black), and 16(green) repeat cTALEs. Units of the bimolecular binding rate constant are $nM^{-1}s^{-1}$, other unimolecular rate constants have units $s^{-1}$. 67.4% confidence intervals are estimated using the conf_interval function of lmfit by performing F-tests (**Newville et al., 2014**). Conditions: 20 mM Tris pH 8.0, 200 mM KCl.

DOI: https://doi.org/10.7554/eLife.38298.014

The following figure supplements are available for figure 5:

**Figure supplement 1.** Colocalization trajectories show TALE-DNA binding and unbinding events.
DOI: https://doi.org/10.7554/eLife.38298.015

**Figure supplement 2.** Bound and unbound lifetimes of 16- and 12-repeat TALE proteins are consistent with multiphasic binding and unbinding.
DOI: https://doi.org/10.7554/eLife.38298.016

**Figure supplement 3.** Distance estimates between labeling sites for $NcTALE_8$ and $NcTALE_{16}$ and the 5' ends of bound DNA.
DOI: https://doi.org/10.7554/eLife.38298.017

Based on this mechanism, the rate laws for binding are given in *Equations 1a - 1d*.

$$\frac{d[TALE]}{dt} = -k_1[TALE] + k_{-1}[TALE*] \tag{1a}$$

$$\frac{d[TALE*]}{dt} = k_1[TALE] - k_{-1}[TALE*] - k_2[TALE*][DNA] \tag{1b}$$

$$\frac{d[TALE* \sim DNA]}{dt} = k_2[TALE*][DNA] \tag{1c}$$

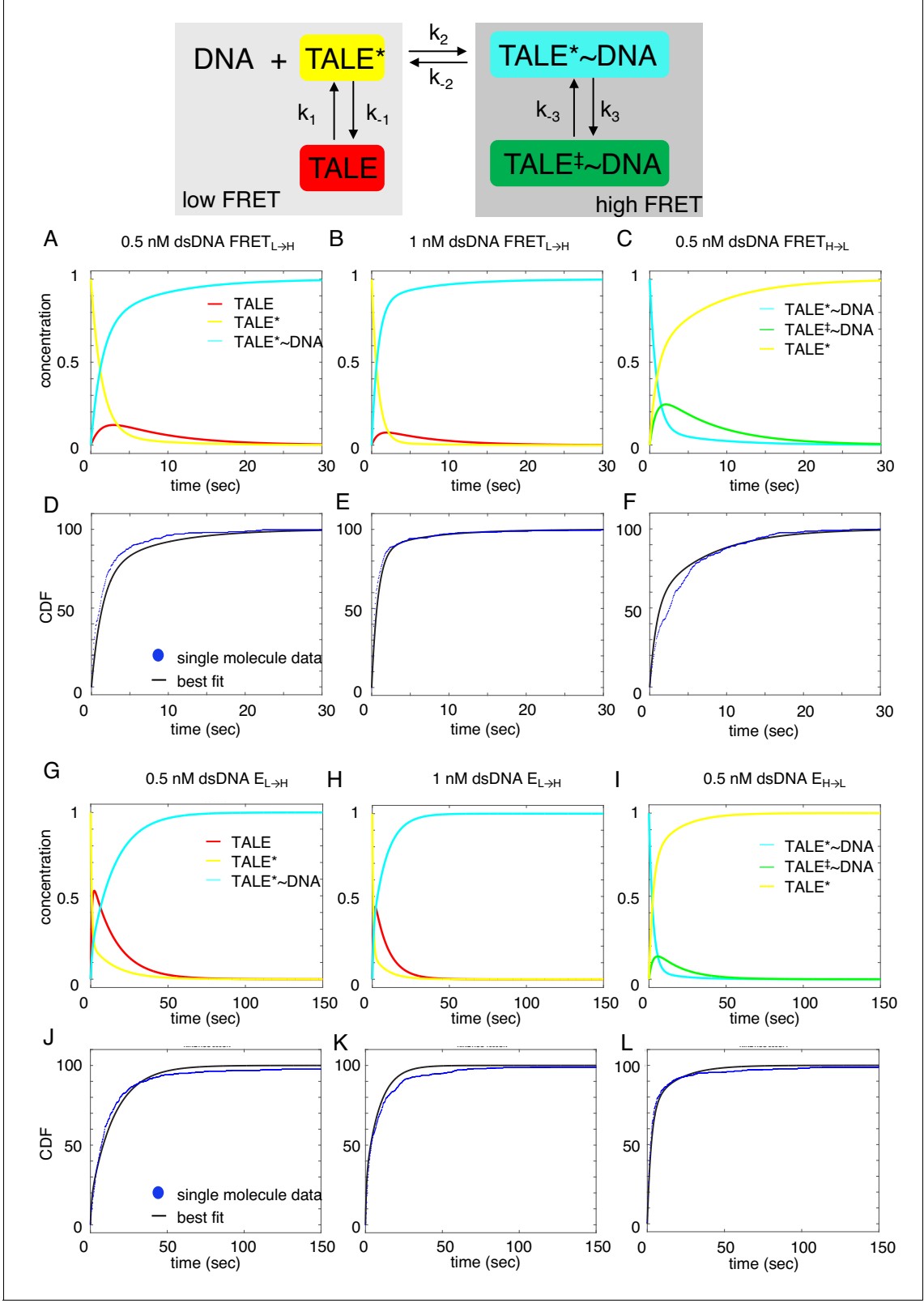

**Figure 6.** Deterministic simulations provide evidence for conformational heterogeneity in the unbound state. The model most consistent with data is shown at the top. Unbound TALEs can exist in DNA-binding competent (TALE*) or DNA-binding incompetent (TALE) states. DNA-bound TALEs can exist in short-lived (TALE*~DNA) or long-lived (TALE‡~DNA) DNA-bound states. Cumulative distributions of dwell-times (shown as blue points) from eight repeat single-molecule time trajectories (A–F) and 16 repeat single-molecule time trajectories (G–L) were analyzed with the model (best-fit shown

*Figure 6 continued on next page*

*Figure 6 continued*

in black). (**A–C and G–I**) Populations of states as a function of time, generated by numerical integration in Matlab. (**D–F and J–L**) Best-fit microscopic rate constants and 68% confidence intervals are listed in *Table 1*.

DOI: https://doi.org/10.7554/eLife.38298.018

$$K_{eq,DNA-free} = \frac{k_1}{k_{-1}} \tag{1d}$$

Since the single-molecule dwell-time histograms of the unbound states are insensitive to the isomerization after DNA binding, the equation describing the time evolution of the long-lived bound state (TALE$^{\ddagger}$~DNA) is not relevant to our analysis of unbound-state lifetimes.

To determine microscopic rate constants $k_1$, $k_{-1}$, and $k_2$, *Equations 1a-1c* were numerically integrated in Matlab, and the fraction of TALE*~DNA as a function of time was fitted to the low-FRET cumulative distributions (NcTALE$_8$; *Figure 6D–E*) or to the no colocalization cumulative distributions (NcTALE$_{16}$; *Figure 6J–K*). Microscopic rate constants were adjusted to reduce sum of the squared residuals between the concentration of TALE*~DNA (the direct product of binding) as a function of time and single-molecule cumulative distributions. In both cases, cumulative distributions at different bulk DNA concentrations were fitted globally. Initial fractions of TALE and TALE*~DNA were set to zero, and the initial fraction of TALE* was set to one. Confidence intervals (CI) were estimated by bootstrapping (*Table 1*; mean and 68% CI from 2000 or 8000 bootstrap iterations).

Rate laws for dissociation are given in *Equations 2a - 2d*

$$\frac{d[TALE*\sim DNA]}{dt} = -k_{-2}[TALE*\sim DNA] - k_3[TALE*\sim DNA] + k_{-3}[TALE^{\ddagger}\sim DNA] \tag{2a}$$

$$\frac{d[TALE^{\ddagger}\sim DNA]}{dt} = k_3[TALE*\sim DNA] - k_{-3}[TALE^{\ddagger}\sim DNA] \tag{2b}$$

$$\frac{d[TALE*]}{dt} = k_{-2}[TALE*\sim DNA] \tag{2c}$$

$$K_{eq,DNA-bound} = \frac{k_3}{k_{-3}} \tag{2d}$$

As with the system of equations above (1a-d), the equation describing the time evolution of the binding-incompetent free state (TALE) is not relevant to our analysis of bound-state lifetimes.

To determine microscopic rate constants $k_{-2}$, $k_{-3}$, and $k_3$, *Equations 2a-2c* were numerically integrated in Matlab, and the fraction of TALE* as a function of time was fitted to the high-FRET cumulative distributions (NcTALE$_8$; *Figure 6F*) or to the low colocalization cumulative distributions (NcTALE$_{16}$; *Figure 6L*). Microscopic rate constants were adjusted to reduce sum of the squared residuals between the concentration of TALE* (the direct product of dissociation) as a function of

**Table 1.** Kinetic parameters obtained from deterministic simulation fits of NcTALEs binding to homopolymeric A/T duplex DNA.

| | $k_1$ (sec$^{-1}$) | $k_{-1}$ (sec$^{-1}$) | $K_{eq,\ DNA-free}$ | $k_2$(sec$^{-1}$nM$^{-1}$) | $k_{-2}$ (sec$^{-1}$) | $k_3$ (sec$^{-1}$) | $k_{-3}$ (sec$^{-1}$) | $K_{eq,\ DNA-bound}$ (nM$^{-1}$) |
|---|---|---|---|---|---|---|---|---|
| NcTALE$_8$[a] | 0.17 [0.16, 0.18] | 0.13 [0.12, 0.14] | 1.32 [1.26, 1.39] | 1.1 [1.08, 1.12] | 0.66 [0.65, 0.67] | 0.36 [0.35, 0.37] | 0.222 [0.218, 0.227] | 1.62 [1.58, 1.66] |
| NcTALE$_{12}$ [b] | 0.135 [0.133, 0.137] | 1.26 [1.09, 1.34] | 0.11 [0.10, 0.12] | 0.31 [0.28, 0.33] | 0.130 [0.129, 0.131] | 0.043 [0.042, 0.044] | 0.0435 [0.0428, 0.0442] | 0.99 [0.97, 1.00] |
| NcTALE$_{16}$ [a] | 0.26 [0.25, 0.27] | 0.43 [0.39, 0.47] | 0.61 [0.57, 0.64] | 0.39 [0.38, 0.41] | 0.299 [0.298, 0.300] | 0.074 [0.073, 0.076] | 0.078 [0.076, 0.079] | 0.96 [0.95, 0.97] |

Parameters for NcTALE$_8$ are for binding to dA$_{15}$/T$_{15}$ duplex DNA; those for NcTALE$_{12}$ and NcTALE$_{16}$ are for binding to dA$_{23}$/T$_{23}$ duplex DNA. 68% confidence intervals shown in brackets are from 2000[a] and 8000[b] iterations of bootstrap analysis.

DOI: https://doi.org/10.7554/eLife.38298.019

time and single-molecule cumulative distributions. In both cases, cumulative distributions at different bulk DNA concentrations were fitted globally. The initial fraction of TALE*~DNA conformer was set at one; all other initial fractions were set to zero. Confidence intervals were estimated by bootstrapping (*Table 1*; mean and 68% CI from 2000 iterations).

Fitted curves reproduce the experimental cumulative distributions for binding and unbinding (*Figure 6*), both for the short and long cTALE arrays, with reasonably small residuals, over a range of DNA concentrations. Generally, fitted rate constants have confidence intervals of 10% or smaller (*Table 1*).

Comparison of microscopic rate constants for 8, 12, and 16 repeats show some significant differences. The bimolecular microscopic binding rate constant, $k_2$, is slightly larger for eight repeats than for 12 and 16 repeats (1.1, 0.31, and 0.39 $nM^{-1}s^{-1}$ for 8, 12, and 16 repeats respectively). However, microscopic unbinding rate constant, $k_{-2}$, is higher for eight repeat cTALEs (0.66 $s^{-1}$ for $NcTALE_8$ versus 0.13 $s^{-1}$ for $NcTALE_{12}$ and 0.299 $s^{-1}$ for $NcTALE_{16}$). *Clauß et al. (2017)* have also observed TALE dissociation rates that are non-monotonic with repeat number in live cells. In addition, bound state isomerization (interconversion between TALE*~DNA and TALE‡~DNA) is 5–10 times slower for 16 and 12 repeat cTALEs than eight repeat cTALEs. The value of $K_{eq, DNA-free}$, which is a measure of the equilibrium proportion of the unbound TALE that is DNA-binding competent (TALE*) to that which is binding-incompetent (TALE), is larger for cTALEs with eight repeats ($K_{eq, DNA-free}$ = 1.32) than for cTALEs with 12 and 16 repeats ($K_{eq, DNA-free}$ = 0.11 and $K_{eq, DNA-free}$ = 0.61, respectively).

## Discussion

By measuring DNA-binding kinetics of cTALE arrays that form 0.7, 1, and 1.4 superhelical turns, we probe the functional relevance of locally unfolded TALE states. We describe a novel method to glean mechanistic details from complex single molecule kinetics. In our simplified cTALE system, we find conformational heterogeneity in both DNA free and DNA-bound states. We find that association is slowed in arrays containing one full turn of repeats or more. Because most natural and designed TALEs contain more than a full turn of repeats, these findings motivate future studies of TALE nucleases (TALEN) to test whether the placement of destabilized repeats at specific positions can increase binding affinity and possibly enhance activity.

### cTALEs containing NS RVD bind DNA with high affinity

NS is an uncommon RVD in natural TALEs. Previous reports suggest that NS is fairly nonspecific, but may bind with higher affinity than other common RVDs (NG, NI, NN, and HD) (*Miller et al., 2015*). Our fitted rate constants can be used to calculate the apparent $K_d$ ($K_{app}$) calculated as follows:

$$K_{app} = \frac{[TALE-DNA+TALE*-DNA]}{[DNA][TALE+TALE*]}$$

$$= \frac{K_{eq,DNA-free}K_2+K_{eq,DNA-free}K_2K_{eq,DNA-bound}}{1+K_{eq,DNA-free}}$$

$$= \frac{\frac{k_1}{k_{-1}}\times\frac{k_2}{k_{-2}}\left(1+\frac{k_3}{k_{-3}}\right)}{1+\frac{k_1}{k_{-1}}}$$

(3)

where $K_2 = k_2/k_{-2}$.

Using fitted rate constants from *Table 1* in the final equality in *Equation 3* gives values for $K_{app}$ of 2.5 nM for the eight repeat cTALE array, 0.5 nM for the 12 repeat cTALE array, and 1.0 nM for the 16 repeat cTALE array. Increasing the number of repeats from 8 to 12 repeats decreases the apparent $K_d$ modestly, but further increasing from 12 to 16 repeats leaves the $K_d$ unchanged. This affinity increase is small compared to that reported in a previous report studying length dependence on the affinity of designed TALEs (dTALEs) (*Rinaldi et al., 2017*), although in that study affinity also became insensitive to repeat number for large arrays when $K_{D,app}$ was in the low nM range. Because the $K_{D,app}$ of $cTALE_8$ is already in the low nM regime (2.5 nM), we speculate that this may represent a similar maximum binding affinity observed for the longer arrays in the previous Rinaldi *et al.* report. Thus, because $cTALE_8$ is near a maximum affinity, the addition of four and eight cTALE repeats has only a modest impact on the apparent binding affinity.

TALEs are believed to read out sequence information from one strand (*Boch et al., 2009*). Due to the asymmetry of our DNA sequences (poly-dA base-paired with poly-dT), in principle, the FRET efficiency contains information on the binding orientation (and thus strand preference). However, based on the crystal structure of the DNA-bound state of TAL-effector PthXo1 (*Mak et al., 2012*), we estimate that the distance between the donor site of NcTALE$_8$ (repeat 1) to the 5' acceptor site on the DNA (Cy5-A$_{15}$/T$_{15}$) should be similar for both the dA-sense or dT-sense orientations (*Figure 5—figure supplement 3A*). Thus, the FRET data does not discriminate between the two modes of binding for the eight-repeat construct. However, for the 16 repeat NS RVD cTALE arrays, the PthXo1 model suggests very different distances (25 Å versus 73 Å for the dT-sense or dA-sense respectively, *Figure 5—figure supplement 3B*) between the donor site (TALE repeat 14) and the acceptor site (5' Cy5-A$_{23}$/T$_{23}$). To restrict the number of binding positions available to longer cTALE arrays, the 23 base pair DNA used for NcTALE$_{16}$ measurements (as well as DNA depicted in *Figure 5—figure supplement 3B*) has the same number of additional base pairs as repeats (eight additional repeats and eight additional base pairs) compared to the 15 base pair DNA used for NcTALE$_8$ measurements (as well as DNA depicted in *Figure 5—figure supplement 3A*). While we limited the number of available binding positions, it may be possible for cTALEs to slide along DNA. However, taking into account the four repeat N-terminal capping domain, there are only three available base pairs in the bound complex. Thus we don't expect the distance measurements to change by more than 10 Å (~3 base pairs) if sliding occurs. The observation that there is colocalization but no measurable FRET when NcTALE$_{16}$ is bound to DNA suggests that cTALEs containing the NS RVD prefer adenine (the dA-sense mode) compared with thymine bases, consistent with previous reports (*Boch et al., 2009*).

## Conformational heterogeneity in the unbound state may be caused by local unfolding

The cumulative distributions of dwell-times in *Figure 3* provide clear evidence for conformational heterogeneity in both the free and DNA-bound cTALEs. Although the deterministic modeling supports such heterogeneity, puts it in the framework of a molecular model, and provides a means to determine the microscopic rate and equilibrium constants, such analysis provides little information about the structural nature of TALE conformational heterogeneity.

*Figure 7* shows a model of cTALE conformational change consistent with DNA binding kinetics. In this model there are four TALE states. DNA-free cTALEs comprise both incompetent and binding competent states. DNA-bound cTALES comprise at least two states that are likely to differ in their registry relative to the DNA. For eight repeat cTALE arrays, the DNA-binding competent state is more highly populated than the DNA-binding incompetent state. In this reaction scheme, the DNA-binding incompetent state can be regarded as an off-pathway conformation that inhibits DNA binding (*Figure 7A*).

Because the eight repeat cTALE array does not form multiple turns of a superhelix, unfolding to bind DNA is not required. In the model in *Figure 7*, the binding competent state is the fully folded conformation, whereas the binding incompetent state includes partly folded conformations. Consistent with this interpretation, increasing populations of partly folded states through addition of 1M urea and through entropy enhancing mutations decreases apparent binding rates of 8 repeat cTALEs (*Figure 7—figure supplement 1*). This is also consistent with a partly folded DNA-binding incompetent state in shorter cTALE arrays.

For 12 and 16 repeat cTALE arrays, the DNA-binding incompetent state is more highly populated than the DNA-binding competent state. In the model in *Figure 7*, the DNA-binding competent state is a high-energy conformation required for DNA binding (*Figure 7B–C*). Because 12 and 16 repeat cTALEs are expected to form 1 and 1.4 turns (excluding the N-terminal domain), we hypothesize that the binding competent state includes some partly folded states that allow access to DNA. Not all partly folded states open the array to access DNA; therefore, the binding incompetent state includes some nonproductive partly folded states in addition to the fully folded state.

In arrays containing 12 or more repeats, the binding competent and binding incompetent states likely include mixtures of many specific partly folded states. Because the types of partly folded states are unknown, connecting equilibria between binding competent and binding incompetent states to calculated partly folded equilibria (using folding free energies similar to *Figure 1*) is challenging. Future work towards understanding the structural characteristics of the binding competent state in TALE arrays of one or more turns would inform which partly folded states to include in the

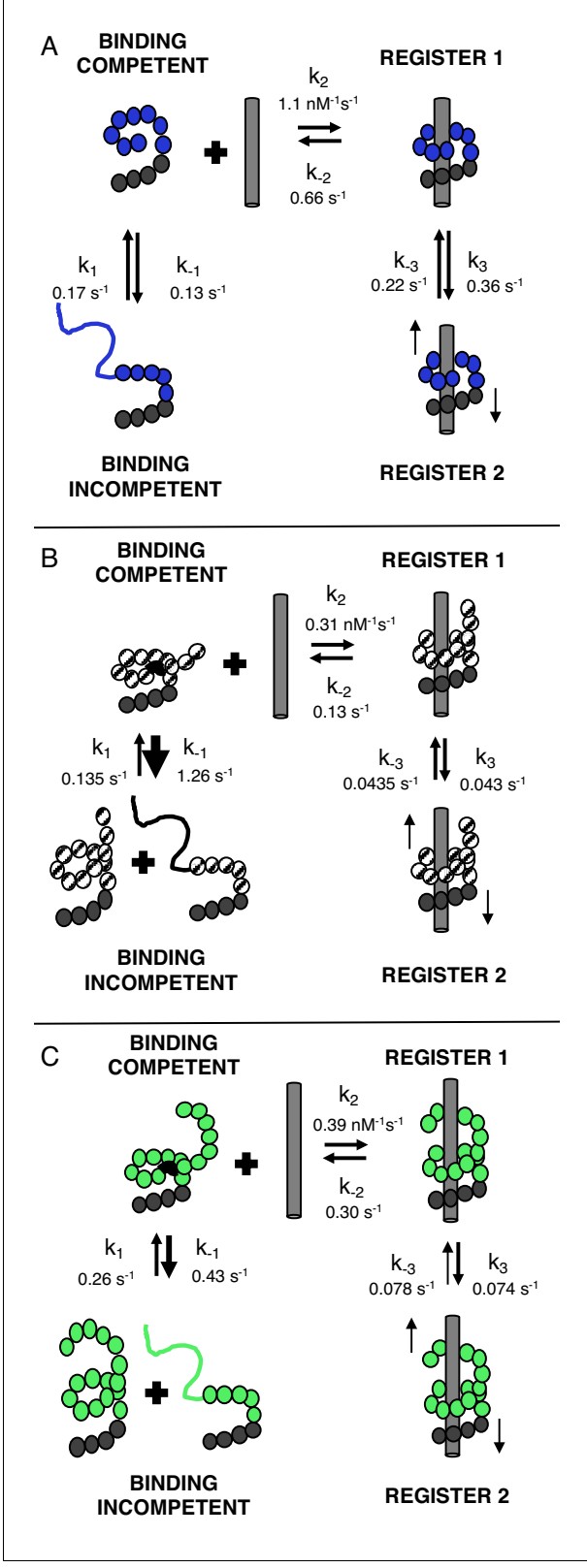

**Figure 7.** TALEs with multiple superhelical turns must break to bind DNA. Single-molecule FRET studies and deterministic modeling support a model where TALEs exist in four states: binding incompetent, binding competent, and bound states in (at least) two distinct registers. In this model, for TALEs that form less than one full superhelical turn (eight repeats, (**A**), partly folded states are off-pathway and slow down binding. For longer

*Figure 7 continued on next page*

*Figure 7 continued*

TALEs that form one (12, **B**) or more (16, **C**) complete superhelical turns, partial unfolding is required for binding. DNA-bound TALEs populate multiple registers with distinct dissociation kinetics. Dynamics of long (12 and 16-repeat; **B–C**) TALEs bound to DNA are significantly slower than for the shorter (8-repeat; **A**) TALEs.
DOI: https://doi.org/10.7554/eLife.38298.020

The following figure supplements are available for figure 7:

**Figure supplement 1.** Urea and destabilizing mutations decrease apparent binding rate of $cTALE_8$.
DOI: https://doi.org/10.7554/eLife.38298.021

**Figure supplement 2.** Deterministic simulations provide evidence for the impact of T-anchored oligo on unbinding kinetics of $cTALE_8$.
DOI: https://doi.org/10.7554/eLife.38298.022

**Figure supplement 3.** Single molecule kinetics in the presence of magnesium cation show a single unbinding phase.
DOI: https://doi.org/10.7554/eLife.38298.023

**Figure supplement 4.** Deterministic simulations provide evidence for the impact of divalent cations on unbinding kinetics of $cTALE_8$.
DOI: https://doi.org/10.7554/eLife.38298.024

calculation, making this comparison meaningful. A better structural understanding of the DNA binding competent state may also allow an opportunity for precise placement of destabilized repeats in designed TALEN arrays which may enable more efficient gene editing methodologies in both clinical and basic research applications.

## TALE functional instability presents a new mode of transcription factor binding

Here we demonstrate kinetic heterogeneity in DNA-bound and unbound TALE arrays, and we subsequently link the observed heterogeneity to partial unfolding of TALE arrays. We propose a model where binding requires partial unfolding of TALE arrays longer than one superhelical turn providing a functional role for previously observed moderate stability of TALE arrays. The functional instability described is particularly surprising given the small population of partly folded states which we expect to be DNA binding competent (partly folded states similar to internally unfolded and interfacially fractured states depicted in *Figure 1A*). Discovery of a functional role for the observed conformational heterogeneity is even more surprising, given the sequence identity of each of our repeats. Sequence heterogeneity in naturally occurring TALE arrays may further enable access to partly folded binding-competent states.

While it is well understood that many transcription factors sometimes undergo local folding transition upon DNA binding (*Spolar and Record, 1994*; *Tsafou et al., 2018*), the findings here indicate that for TALE arrays, the major conformer is fully folded, and must undergo a local unfolding transition in order to bind DNA. Taken together, these findings suggest a new mode of transcription factor binding and provide compelling evidence for functional instability in TALE arrays.

## Conformational heterogeneity in the bound state

Previous reports show that TALEs have multiple diffusional modes when searching nonspecific DNA (*Cuculis et al., 2015*). Our work with the homopolymeric DNA sequences suggests that cTALEs have multiple bound states (*Figure 7*). To gain more insight into conformational heterogeneity in the bound state, we performed binding experiments with T-anchored binding sequences and with divalent magnesium cation (*Figure 4* and *Figure 7—figure supplementa 2–4*). The significant changes in unbinding kinetics suggest that the two kinetically distinct bound-states (TALE*-DNA and TALE-DNA in *Figure 6*) differ in the registry of the TALE-DNA complex. Although we have no structural information on how these two registers differ (for $NcTALE_8$ binding to $A_{15}/T_{15}$ DNA, the two registers appear to have the same FRET efficiency), our deterministic modeling suggests that the two registers differ in their ability to dissociate. The TALE*-DNA state, which we refer to as 'register 1' in the mechanistic model in *Figure 7*, can directly dissociate to the unbound state; likewise, it appears to be the direct product of association. In contrast, the $TALE^{\ddagger}$-DNA state, which we refer to as

'register 2' in *Figure 7*, does not directly dissociate; rather, dissociation from register two involves conversion back to register 1.

To determine how the observed kinetic changes are partitioned into underlying kinetic steps in unbinding, we fitted various kinetic models to the cumulative distributions for unbinding (binding to T-anchored DNA is shown in *Figure 7—figure supplement 2*; binding in the presence of 40 mM $MgCl_2$ is shown in *Figure 7—figure supplement 4*). All unbinding distributions were best-fit by the three-state unbinding model shown in *Figure 5*, yielding lower chi-squared values than fits to a single-exponential model (*Table 2*). Distributions of the best-fit parameters obtained after 2000 bootstrap iterations are normally distributed with small confidence intervals (*Table 2*).

Comparison of the microscopic rate constants for the homopolymeric DNA and T-anchored DNA show some significant differences. The addition of a T-anchor to the binding DNA sequence substantially decreases the rate constant for conversion from register 1 to register 2 ($k_3$, from 0.36 $s^{-1}$ to 0.06 $s^{-1}$, *Figure 7—figure supplement 2* and *Table 2*), and modestly decreases the rate constant for conversion from register 2 to register 1 ($k_{-3}$, from 0.22 $s^{-1}$ to 0.31 $s^{-1}$). The T-anchor also modestly decreases the unbinding rate constant ($k_{-2}$, from 0.66 $s^{-1}$ to 0.48 $s^{-1}$). Taken together, the rate constants from deterministic fits indicate that the addition of T to the binding DNA sequence stabilizes the bound register one state relative to the unbound and register two states.

Comparisons of the microscopic rate constants for cTALE binding to $A_{15}/T_{15}$ DNA in the presence of monovalent $K^+$ and $Mg^{2+}$ also show some significant differences. With $Mg^{2+}$, the unbinding rate constant is larger ($k_{-2}$ = 1.28 $s^{-1}$ vs. 0.66 $s^{-1}$ with $K^+$), as is the rate constant for conversion from register 2 to register 1 ($k_{-3}$ = 1.40 $s^{-1}$ vs. 0.222 $s^{-1}$ with $K^+$); *Figure 7—figure supplement 4* and *Table 2*).

*Table 1* shows that microscopic rate constants for transition between the bound register 1 and register two states become much slower in 12 and 16 repeat cTALEs compared with eight repeat cTALEs ($k_{-3}$ and $k_3$). These rate constants decrease much more than the microscopic unbinding rate constant (the $k_{-2}$ values are 0.66 $s^{-1}$, 0.13 $s^{-1}$, and 0.30 $s^{-1}$ for $NcTALE_8$, $NcTALE_{12}$, and $NcTALE_{16}$ respectively) indicating that the rates of register shifting depend on the number of repeats. Although the model does not provide information on the structure of this conformational change, it is likely that this conformational change involves cTALEs shifting register by 1–3 base pairs on the homopolymeric DNA. Overall, the dissociation results demonstrate that TALE-DNA complexes are heterogeneous, and their rates of interconversion and dissociation depend on sequence, repeat number, and solution conditions.

# Materials and methods

## Cloning, expression, purification, and labeling

Consensus TALE repeat constructs were cloned with C-terminal $His_6$ tags via an in-house version of Golden Gate cloning (*Cermak et al., 2011*). TALE constructs were grown in BL21(T1R) cells at 37°C to an OD of 0.6–0.8 and induced with 1 mM IPTG. Following cell pelleting and lysis, proteins were purified by resuspending the insoluble material in 6M urea, 300 mM NaCl, 0.5 mM TCEP, and 10 mM $NaPO_4$ pH 7.4. Constructs were loaded onto a Ni-NTA column. Protein was eluted using 250 mM imidazole and refolded during buffer exchange into 300 mM NaCl, 30% glycerol, 0.5 mM TCEP, and 10 mM $NaPO_4$ pH 7.4.

**Table 2.** $NcTALE_8$ kinetic unbinding parameters obtained from deterministic fits.

| | DNA | Salt | $k_{-2}$ (sec$^{-1}$) | $k_3$ (sec$^{-1}$) | $k_{-3}$ (sec$^{-1}$) | $K_{eq, \text{ DNA-bound}}$ (nM$^{-1}$) |
|---|---|---|---|---|---|---|
| 1 | Cy5-$A_{15}/T_{15}$ dsDNA | 200 mM KCl | 0.66 [0.65, 0.67] | 0.36 [0.35, 0.37] | 0.222 [0.218, 0.227] | 1.62 [1.58, 1.66] |
| 2 | Cy5-$TA_{14}/T_{14}A$ dsDNA | 200 mM KCl | 0.48 [0.475, 0.483] | 0.06 [0.054, 0.068] | 0.31 [0.28, 0.33] | 0.199 [0.192, 0.205] |
| 3 | Cy5-$A_{15}/T_{15}$ dsDNA | 40 mM $MgCl_2$ | 1.28 [1.27, 1.30] | 0.32 [0.28, 0.35] | 1.40 [1.30, 1.51] | 0.224 [0.211, 0.237] |

Mean and 68% confidence intervals shown in brackets are from 2000 iterations of bootstrap analysis.
DOI: https://doi.org/10.7554/eLife.38298.025

Labelling of cTALE arrays followed a previously reported protocol (*Rasnik et al., 2004*). NcTALE$_8$ and NcTALE$_{12}$ were labeled at residue R30C in the first repeat, while NcTALE$_{16}$ was labeled at residue R30C in the fourteenth repeat. 1 mg protein was loaded onto 500 uL NiNTA spin column. The column as washed with 10 column volumes of 300 mM NaCl, 0.5 mM TCEP, and 10 mM NaPO$_4$ pH 7.4. Tenfold molar excess Cy3 maleimide dye was resuspended in 10 μL DMSO and added to column. The column was rocked at room temperature for 30 min, then at 4°C overnight. Cy3-labeled protein was eluted with 250 mM imidazole, 300 mM NaCl, 30% glycerol, 0.5 mM TCEP, and 10 mM NaPO$_4$ pH 7.4. Protein was stored in 300 mM NaCl, 30% glycerol, 0.5 mM TCEP, and 10 mM NaPO$_4$ pH 7.4 at −80°C.

## Calculation of partly folded state free energies

Free energies of partly folded cTALE conformations were determined using previously reported TALE intrinsic and interfacial free energies (*Geiger-Schuller and Barrick, 2016*). Free energies in *Figure 1* are difference between the partly and fully folded states (where all 20 repeats are folded with coupled interfaces). Previously determined intrinsic and interfacial free energies were used to calculate probabilities of the fully folded state (all repeats folded), the end-frayed state (the first of twenty repeats unfolded), the internally unfolded state (the tenth of twenty repeats unfolded), and the interfacially fractured state (the interface between repeat ten and eleven disrupted). As an example, to calculate the free energy of end-fraying, the end-frayed state probability is divided by the fully folded state probability to generate the equilibrium constant for end-fraying (K$_{end-frayed}$), which is used to calculate the free energy of end-fraying:

$$\Delta G_{end-frayed} = -RT \ln K_{end-frayed} \tag{4}$$

The conceptual framework and mathematical description of the Ising model and folding free energies are described in *Aksel and Barrick (2009)*.

## Oligonucleotides

Sequences used for binding studies were 5'-Cy5-A$_{15}$-3' and 5' T$_{15}$-3' duplex (Cy5-A$_{15}$/T$_{15}$) as well as 5'-Cy5-A$_{15}$-3' and 5'-biotin-T$_{15}$-3' duplex (Cy5-A$_{15}$/biotin-T$_{15}$) for eight repeat binding studies, and 5'-Cy5-A$_{23}$-3' and 5' T$_{23}$-3' duplex (Cy5-A$_{23}$/T$_{23}$) for 12 and 16 repeat binding studies. Sequences used for T-anchored binding studies were 5'-Cy5-TA$_{14}$-3' and 5' T$_{14}$A-3' duplex (Cy5-TA$_{14}$/T$_{14}$A) for eight repeat binding studies. DNA was annealed at 5 μM concentration with 1.2-fold molar excess unlabeled strand in 10 mM Tris pH 7.0, 30 mM NaCl.

## Single-molecule detection and data analysis

Biotinylated quartz slides and glass coverslips were prepared as previously described (*Rasnik et al., 2004*). Cy3-labeled cTALEs were immobilized on biotinylated slides taking advantage of neutravidin interaction with biotinylated α-penta•His antibody which binds the His$_6$ cTALE tag. Slides were pretreated with blocking buffer (5 μL yeast tRNA, 5 μL BSA, 40 μL T50) before addition of 250 pM labeled cTALE. Cy5-labeled duplex DNA was mixed with imaging buffer (20 mM Tris pH 8.0, 200 mM KCl, 0.5 mg mL$^{-1}$ BSA, 1 mg mL$^{-1}$ glucose oxidase, 0.004 mg mL$^{-1}$ catalase, 0.8% dextrose and saturated Trolox ~1 mg mL$^{-1}$) and molecules were imagined using total internal reflection fluorescence microscopy. The time resolution was 50 msec for NcTALE$_8$ and 100 msec for NcTALE$_{16}$ and NcTALE$_{12}$. Collection and analysis was performed as previously described (*Roy et al., 2008*).

## FRET histograms

A minimum of 20 short movies were collected, and the first five frames (50 msec exposure time) were used to generate smFRET histograms. FRET was calculated as I$_A$/(I$_A$ +I$_D$) where I$_A$ and I$_D$ are donor-leakage and background corrected fluorescence emission of acceptor (Cy5) and donor (Cy3) fluorophores. In competition experiments, unlabeled DNA with the same sequence as labeled DNA was mixed at indicated concentrations with labeled DNA prior to imaging.

## Dwell time analysis

Long movies were collected with 50 msec exposure time for NcTALE$_8$ and 100 msec exposure time for NcTALE$_{16}$ and NcTALE$_{16}$. At least 20 representative traces at each DNA concentration were

selected and dwell times were determined by fitting as previously described using HaMMy (*McKinney et al., 2006*) for FRET in NcTALE$_8$. Dwell times in NcTALE$_{12}$ and NcTALE$_{16}$ colocalization experiments are determined by using a thresholding procedure for Cy5 excitation (*Figure 5—figure supplement 1*). The algorithm used to identify low and high emission states here is slightly different than previously described thresholding algorithms (*Blanco and Walter, 2010*). To reduce the number of incorrectly identified transitions arising from increased background and noise at higher Cy5-labeled DNA concentrations, a thresholding algorithm with two limits was implemented (see *Figure 5—figure supplement 1*). All FRET and colocalization data are well described by models with two distinct states (0.0 FRET and ~0.45 FRET as well as low colocalization and high colocalization). Dwell times of the same state (low versus high FRET or low versus high colocalization) for all traces at a given DNA concentration are compiled, and cumulative distribution is generated with spacing equal to imaging exposure time.

To determine apparent rate constants using model-independent analysis, cumulative distributions were fitted with single and double exponential decays (*Figures 3* and *4*). Observed rates from exponential decay fits were plotted as a function of DNA concentration. Apparent rate constants were calculated as slope of DNA concentration-dependent observed rates or average of DNA concentration-independent observed rates.

## Deterministic modeling

*Equations 1a-1c and 2a-2c* were numerically integrated using the ODE15s and ODE45 solver in MATLAB. Microscopic rate constants were adjusted to minimize the sum of squared residuals between ODE-determined concentration of bound or free TALE and single molecule cumulative distributions using lsqnonlin in MATLAB. 68% confidence intervals were estimated by performing 2000 or 8000 bootstrap iterations in which residuals from the best fit of the model to the data were randomly re-sampled (with replacement) and re-fitted. All scripts and source data required to run this MATLAB program called **De**terminstic **M**odeling for **A**nalysis of complex **S**ingle molecule **K**inetics (DeMASK) are publicly available on GitHub at https://github.com/kgeigers/DeMASK (*Geiger-Schuller, 2019*; copy archived at https://github.com/elifesciences-publications/DeMASK).

## Acknowledgements

The authors thank members of the Barrick and Ha lab for their input on this work. The authors acknowledge the support of the Center for Molecular Biophysics at Johns Hopkins and Dr. Katherine Tripp for instrumental and technical support. Support to KGS was provided by NIH training grant T32-GM008403. Support for this project was provided by NIH grant 1R01-GM068462 to DB and GM112659 to TH and NSF grant PHY 1430124 to TH.

## Additional information

### Competing interests

Taekjip Ha: Reviewing editor, *eLife*. The other authors declare that no competing interests exist.

### Funding

| Funder | Grant reference number | Author |
| --- | --- | --- |
| National Institute of General Medical Sciences | T32-GM008403 | Kathryn Geiger-Schuller |
| National Institute of General Medical Sciences | GM1129659 | Taekjip Ha |
| National Science Foundation | PHY 1430124 | Taekjip Ha |
| National Institute of General Medical Sciences | R01-GM068462 | Doug Barrick |

The funders had no role in study design, data collection and interpretation, or the decision to submit the work for publication.

## Author contributions
Kathryn Geiger-Schuller, Conceptualization, Software, Formal analysis, Investigation, Writing—original draft, Writing—review and editing; Jaba Mitra, Conceptualization, Investigation, Writing—review and editing; Taekjip Ha, Doug Barrick, Conceptualization, Supervision, Writing—review and editing

## Author ORCIDs
Kathryn Geiger-Schuller http://orcid.org/0000-0002-6705-0681
Taekjip Ha https://orcid.org/0000-0003-2195-6258
Doug Barrick http://orcid.org/0000-0001-7291-1389

## Decision letter and Author response
Decision letter https://doi.org/10.7554/eLife.38298.030
Author response https://doi.org/10.7554/eLife.38298.031

# Additional files

## Supplementary files
• Transparent reporting form
DOI: https://doi.org/10.7554/eLife.38298.026

## Data availability
Source data files have been provided for Figure 3. Source code and data files related to Figure 6 are publicly available and can be found at https://github.com/kgeigers/DeMASK (copy archived at https://github.com/elifesciences-publications/DeMASK) and on Zenodo (http://doi.org/10.5281/zenodo.2538666).

The following dataset was generated:

| Author(s) | Year | Dataset title | Dataset URL | Database and Identifier |
|---|---|---|---|---|
| Geiger-Schuller KR | 2019 | Source code for DeMASK: Deterministic Modeling for Analysis of complex Single molecule Kinetics | http://doi.org/10.5281/zenodo.2538666 | Zenodo, 10.5281/zenodo.2538666 |

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
