## [Decision Letter]

Thank you for submitting your article "Functional instability allows access to DNA in longer Transcription Activator-Like Effector (TALE) arrays" for consideration by *eLife*. Your article has been reviewed by John Kuriyan as the Senior Editor, a Reviewing Editor, and two reviewers. The following individual involved in review of your submission has agreed to reveal his identity: Adam J Bogdanove.

The reviewers have discussed the reviews with one another and the Reviewing Editor has drafted this decision to help you prepare a revised submission. While the reviewers are positive about the relevance of the problem and the potential importance of the results in understanding how partially folded states help TALE binding, they raise a number of significant concerns. Each of these concerns can be addressed by performing additional experiments.

Summary:

In this manuscript, the authors use a combination of single-molecule FRET experiments and modeling to study the kinetics of DNA binding for TALE constructs containing different length repeat sequences (8, 12, 16 repeats). The main results include evidence of conformational heterogeneity of the DNA-bound state for TALE proteins, together with the development of a kinetic model that explains the role of different interconvertible binding-competent and binding-incompetent states in DNA binding and how the largely fully folded protein must undergo a local unfolding transition to bind DNA. The experimental approach and the modeling are elegant and widely applicable and therefore likely of broad interest.

Essential revisions:

1) By using a homopolymeric cTALE (all NS) and a homopolymeric substrate (all A on one strand and all T on the other), the experimental design provides no anchor for register. As an alternative to the hypothesis presented in Figure 6, it is possible that the partially bound ("encounter") state inferred by the authors is due to entry somewhere in the middle of the DNA, resulting in some corresponding C-terminal number of repeats "hanging off" the end (or to the side). In such a scenario, couldn't the slower transition of the longer cTALEs to the "locked" (fully bound) state (the bound state isomerization, subsection “A deterministic approach to modeling cTALE-DNA binding kinetics”) be the result of the longer time need to slide 5' along the DNA to allow the C-terminal repeats to hop or fold on (or could the "loose" end be contributing energy toward dissociation, or some combination)? The authors should repeat the experiments using some type of anchor for register. TALEs have a strong preference for T at position -1 in the binding site, dictated by the N-terminal cryptic repeats, and these repeats provide much of the binding energy. Indeed, they are theorized to nucleate the protein-DNA interaction. Incorporating a T at the beginning of the polyA sequence should do the trick, but for good measure, it wouldn't hurt to also incorporate a few different, specific RVDs and corresponding bases early in the arrays. Based on the available structures, RVDs do not appear to differentially impact the inter-repeat interfaces.

Perhaps related to the potential problem of no anchor, it is surprising that the 16 repeat cTALEs do not have significantly higher affinity than the 8 repeat cTALE and that the relationship between length and the bimolecular microscopic binding and unbinding constants is not linear (subsection “A deterministic approach to modeling cTALE-DNA binding kinetics”). Rinaldi et al., showed that increasing numbers of repeats increases affinity for target DNA but that the gain in affinity with more repeats decays exponentially (they observed that affinity for non-specific DNA increases as well, but with a slower rate of decay of gain). Despite the decrease in the rate of gain, one would still expect to see higher affinity for the 12 and 16 repeat cTALEs relative to the 8. Is it possible that, unanchored, cTALE-DNA interactions that do not span the length of the array are confounding the results observed by the authors? The authors surmise based on the estimated distance between the fluorophores (Figure 4—figure supplement 3) that the absence of FRET using the 16 repeat TALE is due to its binding the 'A' sense strand rather than the 'T' sense strand, and they are correct in observing that NS prefers A to T, but NS can bind T (Miller et al., 2015), so it is unclear whether some population of the protein is wrapped in the other direction, using one of the T's as its position -1. As above, the authors should address this possibility in their discussion or with an anchored setup.

2) Single molecule FRET experiments are performed by surface tethering the C-terminus of the repeat region to a solid surface. A major finding of the paper is the TALEs exhibit conformational heterogeneity during dynamics. With the protein immobilized, there is a concern that the surface tethering affects conformational changes and kinetics during these dynamic processes. A critical control experiment is to have the DNA anchored to the surface and the protein free in solution.

3) Most/all of the experiments reported conditions of monovalent salt. Did the authors include any divalent salt in solution during any of the experiments? It would seem that in vivo conditions would include some amount of divalents. The main point is that all of the kinetic parameters were determined with monovalent salts only.

4) FRET was observed for the Cy3-labeled TALE with an 8-repeat sequence, but not for the 12 and 16-repeat TALEs. Why? This is not clear from the paper. How did the authors decide on the location of dye labeling for the TALE proteins? The structure shown in Figure 4—figure supplement 3 (as best as I can tell) suggests that FRET should be possible for the larger TALE constructs. Second, the 8 and 12 repeat TALEs were labeled at the first repeat, whereas the 16 repeat TALE was labeled at the 14th repeat. What is the rationale for this labeling strategy? Clearly labeling location will affect the absence or presence of FRET, and it is unclear that this was studied in a systematic manner.

5) Only the 8 repeat TALE exhibited FRET, whereas the 12 and 16 repeat TALEs did not. For the 8-repeat TALE, a transition to low FRET is interpreted as "likely an unbinding event". However, no FRET is then observed for the 12 and 16 repeat TALEs, so the co-localization method is used. The main point here is that the interpretation of FRET (or acceptor signal) does not seem to be uniform between the different constructs in terms of the actual structural information or dynamics. For example, does this mean that the low FRET state in the 8-repeat TALE is fundamentally different than no FRET in the 12 and 16-repeat sequences? This is related to the next point (6).

6) In drawing schematics such as that shown in Figure 6, and in using the FRET (or localization data) to extract rate constants from the kinetic model, the authors rely on a large amount of speculation about the underlying dynamics and structure (or hypothesis). In this particular case, the behavior in the schematic (Figure 6) does not appear (in my opinion) to be solidly supported by the experimental data presented in the paper. It is largely speculative, and as the authors admit in the Discussion section "such [experimental/computational] analysis provides little information about the structural nature of TALE conformational heterogeneity". How can we directly attribute different kinetic states to partially folded (or not) TALE proteins? Bottom line is that the ideas in the schematic are interesting and potentially novel (and significant), but too preliminary to be proposed based on the experimental data.

7) In the Introduction, the authors refer to 'partly folded or broken states' without sufficient explanation. These ideas are drawn heavily upon in the paper, but they need more clear explanation. Moreover, are the concepts in Figure 1 presented as new results and findings for this paper, or are they presented as prior results? In either case, more explanation is needed. How are δ G values determined?

---

## [Author Response]

Essential revisions:1) By using a homopolymeric cTALE (all NS) and a homopolymeric substrate (all A on one strand and all T on the other), the experimental design provides no anchor for register. As an alternative to the hypothesis presented in Figure 6, it is possible that the partially bound ("encounter") state inferred by the authors is due to entry somewhere in the middle of the DNA, resulting in some corresponding C-terminal number of repeats "hanging off" the end (or to the side). In such a scenario, couldn't the slower transition of the longer cTALEs to the "locked" (fully bound) state (the bound state isomerization, subsection “A deterministic approach to modeling cTALE-DNA binding kinetics”) be the result of the longer time need to slide 5' along the DNA to allow the C-terminal repeats to hop or fold on (or could the "loose" end be contributing energy toward dissociation, or some combination)? The authors should repeat the experiments using some type of anchor for register. TALEs have a strong preference for T at position -1 in the binding site, dictated by the N-terminal cryptic repeats, and these repeats provide much of the binding energy. Indeed, they are theorized to nucleate the protein-DNA interaction. Incorporating a T at the beginning of the polyA sequence should do the trick, but for good measure, it wouldn't hurt to also incorporate a few different, specific RVDs and corresponding bases early in the arrays. Based on the available structures, RVDs do not appear to differentially impact the inter-repeat interfaces.Perhaps related to the potential problem of no anchor, it is surprising that the 16 repeat cTALEs do not have significantly higher affinity than the 8 repeat cTALE and that the relationship between length and the bimolecular microscopic binding and unbinding constants is not linear (subsection “A deterministic approach to modeling cTALE-DNA binding kinetics”). Rinaldi et al., showed that increasing numbers of repeats increases affinity for target DNA but that the gain in affinity with more repeats decays exponentially (they observed that affinity for non-specific DNA increases as well, but with a slower rate of decay of gain). Despite the decrease in the rate of gain, one would still expect to see higher affinity for the 12 and 16 repeat cTALEs relative to the 8. Is it possible that, unanchored, cTALE-DNA interactions that do not span the length of the array are confounding the results observed by the authors? The authors surmise based on the estimated distance between the fluorophores (Figure 4—figure supplement 3) that the absence of FRET using the 16 repeat TALE is due to its binding the 'A' sense strand rather than the 'T' sense strand, and they are correct in observing that NS prefers A to T, but NS can bind T (Miller et al., 2015), so it is unclear whether some population of the protein is wrapped in the other direction, using one of the T's as its position -1. As above, the authors should address this possibility in their discussion or with an anchored setup.

The main criticism of the reviewers (comment 1) is that our DNAs and protein constructs are homopolymeric, and that as a result, it may be that our binding is heterogeneous, with out-of-register binding configurations as well as the intended in-register configuration where each TALE repeat is paired with an AT base. Such a scenario might suggest that the kinetic complexity we see in unbinding results from binding in multiple registers along the DNA. To test this, we performed additional experiments with a T-anchored DNA (Cy5-TA_14_/A_14_T) as suggested. We show these data inthe new Figure 4 and describe them in the new subsection “Modifying the dsDNA sequence to include an anchoring 5’ T impacts unbinding kinetics”.We find that binding to the Tanchored DNA remains multi-phasic (similar to the homopolymeric DNA), but unbinding kinetics are impacted. This result suggests that indeed, heterogenous bound states contribute to complexity in unbinding kinetics.

We discuss this bound state heterogeneity in the subsection “Conformational heterogeneity in the bound state”. We interpret the bound state heterogeneity in the homopolymeric constructs as conversion between distinct isoenergetic states (‘register 1’ and ‘register 2’). With homopolymeric constructs, there are several additional DNA base pairs (1-3 in the case of cTALE_8_ binding A_15_/T_15_) available as alternate binding registries. Without unbinding, cTALEs may shift registry one or more base pairs where all repeats still contact DNA. Additionally, binding should also be able to occur where the protein binds the DNA near to one end, and only binds with a subset of TALE repeats (and bases), although such partly bound states should be higher in energy and as such, should contribute little to the observed kinetics. The Tanchored DNA experiments suggested by the reviewers provided great insight into the unbinding mechanism and to heterogeneity in the bound state. Deterministic analysis shows that there is still some (albeit significantly decreased) heterogeneity in the bound state by breaking the symmetry and stabilizing one of the two registries (in which, we presume, the T-anchor is engaged with the N-terminal cap). Consistent with this stabilization, we detect a significantly slower conversion from ‘registry 1’ to ‘registry 2’ as well as a decrease in the unbinding rate constant in unbinding kinetics measured with the T-anchoring DNA.

In terms of the length dependence of DNA affinity, the reviewers are correct that we do not see much decrease between arrays of 8, 12, and 16 repeats. Table 1 shows a two-fold decrease in Kd going from 8 to 12 repeats, but essentially no decrease going from 12 to 16 repeats. The elegant study of Rinaldi et al., (2017) clearly shows an increase in affinity over this range, although for longer arrays the increase in affinity appears to saturate. Also, Rinaldi et al., show that for two different DNA sequences (specific and nonspecific), saturation occurs at different lengths. It is possible that in our homopolymeric system this saturation is "rightshifted" compared to those of Rinaldi et al., Figure 3 and Figure 4. We share the reviewers' intuition that affinity should increase with length. We speculate that because cTALEs have a high binding affinity at 8 repeats (K_D,app_ = 2.5 nM for cTALE8 which is similar to Rinaldi et al. reported specific binding K_D,app_ for 26 RVDs), adding additional repeats does not increase this affinity very much. We expect that if we looked at cTALEs with fewer repeats we may have seen a relationship with the number of repeats and binding affinity that followed a similar trend as what was observed in the elegant Rinaldi et al. We discuss this in the text added to subsection “cTALEs containing NS RVD bind DNA with high affinity”.

As to potential binding in the opposite orientation, the similar FRET level for the T-anchored TALE complex suggests the two DNAs bind in the same orientation. As we expect the T-anchored DNA to bind in the intended orientation, we expect the orientation is the same for the polyA sequences.

2) Single molecule FRET experiments are performed by surface tethering the C-terminus of the repeat region to a solid surface. A major finding of the paper is the TALEs exhibit conformational heterogeneity during dynamics. With the protein immobilized, there is a concern that the surface tethering affects conformational changes and kinetics during these dynamic processes. A critical control experiment is to have the DNA anchored to the surface and the protein free in solution.

We performed additional experiments with tethered dsDNA binding to cTALEs from bulk solution. We show in Figure 2—figure supplement 1 and the updated subsection “Untethered cTALEs bind dsDNA” that DNA binding is similar in the original orientation (with tethered cTALEs) and the reverse orientation (with tethered dsDNA). Unfortunately, we cannot study a wide range of protein concentrations since the protein aggregates at low salt and low glycerol concentrations.

*3) Most/all of the experiments reported conditions of monovalent salt. Did the authors include any divalent salt in solution during any of the experiments? It would seem that* in vivo *conditions would include some amount of divalents. The main point is that all of the kinetic parameters were determined with monovalent salts only.*

We performed additional experiments in the presence of divalent cations (Mg^2+^). We display these data in Figure 7—figure supplement 3 and Figure 7—figure supplement 4 and discuss them in the updated subsection “Conformational heterogeneity in the bound state”. Similar to what we previously observed with 200 mM KCl, binding and unbinding kinetics are multi-phasic in the presence of 40 mM MgCl2, although we do observe changes in the rate of unbinding as well as the rate of conversion between different isoenergetic bound state registries (referred to here as ‘register 1’ and ‘register 2’).

4) FRET was observed for the Cy3-labeled TALE with an 8-repeat sequence, but not for the 12 and 16-repeat TALEs. Why? This is not clear from the paper. How did the authors decide on the location of dye labeling for the TALE proteins? The structure shown in Figure 4—figure supplement 3 (as best as I can tell) suggests that FRET should be possible for the larger TALE constructs. Second, the 8 and 12 repeat TALEs were labeled at the first repeat, whereas the 16 repeat TALE was labeled at the 14th repeat. What is the rationale for this labeling strategy? Clearly labeling location will affect the absence or presence of FRET, and it is unclear that this was studied in a systematic manner.

We provide a better description of our rationale for attachment of fluorophores in the updated subsection “Longer cTALEs have slower DNA binding and unbinding kinetics”. We observe a low FRET value near 0.2 for the A_23_/T_23_ bound cTALE_12_ state indicating that the fluorophores on cTALE_12_ and A_23_/T_23_ dsDNA are further apart than in cTALE_8_ binding to A_15_/T_15_. Attempts to increase the FRET efficiency by moving the FRET donor label to a different location in the cTALE_16_ protein (to the fourteenth repeat) were unsuccessful. Because we see correlated changes in FRET and colocalization when monitoring kinetics of cTALE_8_ binding and unbinding (Figure 3—figure supplement 1), we feel confident in comparing both FRET and colocalization kinetics.

5) Only the 8 repeat TALE exhibited FRET, whereas the 12 and 16 repeat TALEs did not. For the 8-repeat TALE, a transition to low FRET is interpreted as "likely an unbinding event". However, no FRET is then observed for the 12 and 16 repeat TALEs, so the co-localization method is used. The main point here is that the interpretation of FRET (or acceptor signal) does not seem to be uniform between the different constructs in terms of the actual structural information or dynamics. For example, does this mean that the low FRET state in the 8-repeat TALE is fundamentally different than no FRET in the 12 and 16-repeat sequences? This is related to the next point (6).

Our response to the previous point (4) helps resolve this question. We describe rationale for attachment of fluorophores and decision to measure colocalization of FRET in the updated subsection “Longer cTALEs have slower DNA binding and unbinding kinetics”. We do think there is a difference between the low-FRET "state" for the 12- and 16-repeat constructs, because at high protein concentration, most of the protein is bound to DNA, even though it does not show FRET. We know it is bound from co-localization. We can only speculate on the structural differences that prevent FRET in the bound states with longer constructs, but our main goal here is to use fluorescence as a signal for binding, which the co-localization studies provide.

6) In drawing schematics such as that shown in Figure 6, and in using the FRET (or localization data) to extract rate constants from the kinetic model, the authors rely on a large amount of speculation about the underlying dynamics and structure (or hypothesis). In this particular case, the behavior in the schematic (Figure 6) does not appear (in my opinion) to be solidly supported by the experimental data presented in the paper. It is largely speculative, and as the authors admit in the Discussion section "such [experimental/computational] analysis provides little information about the structural nature of TALE conformational heterogeneity". How can we directly attribute different kinetic states to partially folded (or not) TALE proteins? Bottom line is that the ideas in the schematic are interesting and potentially novel (and significant), but too preliminary to be proposed based on the experimental data.

See Figure 7—figure supplement 1 and updated subsection “Conformational heterogeneity in the bound state”. We think that the experiments with the T-anchored DNA provide additional insight into the heterogeneity in the bound state, demonstrating that the DNA plays a key role in this heterogeneity. Thus, we think we are on more solid ground in our picture of heterogeneity in the bound state relating to shifts in binding register.

As to the multiphasic binding kinetics, there are a few possibilities. One possibility is that either of the reactants (the protein or the DNA) could be heterogeneous (which is our interpretation). In bulk kinetic schemes, heterogeneity can also result from intermediates along a reaction pathway. We know which kinetic phase corresponds to association (the fast one) since it is DNA concentration dependent. In principle, the kinetic intermediate could either before or after the rate-limiting step. But in our case, since we know that the product of the fast phase has full bound-state fluorescence, the "kinetic intermediate" cannot be past the DNA binding step, because we would not be able to detect it. Thus, the intermediate is before the binding step. This is consistent with our data, where the "intermediate" is a binding competent form of the TALE protein. Although it is possible the kinetic complexity in binding comes from the DNA, we expect these simple oligonucleutids to be pretty homogeneous; instead, our previous measurement of the conformational heterogeneity of these TALE arrays through Ising analysis suggests a high level of conformational heterogeneity in the protein, so we feel it is not unreasonable to suggest a connection between these two observations. Also, data shown in Figure 6—figure supplement 1 demonstrate that increasing populations of partly folded states in the 8 repeat construct (through both chemical and mutational destabilization) decreases apparent binding rates, further implicating the protein as the source of heterogeneity. We have more clearly referenced this data in the subsection “Conformational heterogeneity in the bound state” and thoroughly explain the significance of this evidence to support our model in the revised manuscript. We hope that in a revised manuscript, the reviewer will agree.

7) In the Introduction, the authors refer to 'partly folded or broken states' without sufficient explanation. These ideas are drawn heavily upon in the paper, but they need more clear explanation. Moreover, are the concepts in Figure 1 presented as new results and findings for this paper, or are they presented as prior results? In either case, more explanation is needed. How are δ G values determined?

The concepts from Figure 1 are presented as prior results which motivate our hypotheses related to complex binding kinetics, and we modified the text to make this clear. We updated the Introduction and added a new subsection “Calculation of partly folded state free energies” to describe the calculation of these deltaG values.